# Source apportionment and ecotoxicity of PM$_{2.5}$ pollution events in a major Southern Hemisphere megacity: influence of a biofuel-impacted fleet and biomass burning

**Guilherme Martins Pereira**[1,2], **Leonardo Yoshiaki Kamigauti**[1], **Rubens Fabio Pereira**[1],
**Djacinto Monteiro dos Santos**[3], **Thayná da Silva Santos**[2,a], **José Vinicius Martins**[4], **Célia Alves**[5],
**Cátia Gonçalves**[5,b], **Ismael Casotti Rienda**[5], **Nora Kováts**[6], **Thiago Nogueira**[7], **Luciana Rizzo**[3],
**Paulo Artaxo**[3], **Regina Maura de Miranda**[8], **Marcia Akemi Yamasoe**[1], **Edmilson Dias de Freitas**[1],
**Pérola de Castro Vasconcellos**[2], and **Maria de Fatima Andrade**[1]

[1]Department of Atmospheric Sciences, Institute of Astronomy, Geophysics and Atmospheric Sciences,
University of São Paulo, 05508-090, São Paulo, Brazil
[2]Department of Chemistry, Institute of Chemistry, University of São Paulo, 05508-000, São Paulo, Brazil
[3]Department of Applied Physics, Institute of Physics, University of São Paulo, 05508-090, São Paulo, Brazil
[4]Department of Mineralogy and Geotectonics, Institute of Geosciences,
University of São Paulo, 05508-080, São Paulo, Brazil
[5]Centre for Environmental and Marine Studies, Department of Environment and Planning,
University of Aveiro, 3810-193 Aveiro, Portugal
[6]Center of Natural Environmental Sciences, University of Pannonia, Egyetem str. 10, 8200 Veszprém, Hungary
[7]Department of Environmental Health, School of Public Health,
University of São Paulo, 01246-904, São Paulo, Brazil
[8]School of Arts, Sciences and Humanities, University of São Paulo, 03828-000, São Paulo, Brazil
[a]current address: Department of Meteorology, Federal University of Rio de Janeiro, Rio de Janeiro, Brazil
[b]current address: Department of Physics, University of León, Campus de Vegazana, 24071 León, Spain

**Correspondence:** Guilherme Martins Pereira (guilherme.martins.pereira@usp.br)

**Abstract.** The Metropolitan Area of São Paulo (MASP) in Brazil has reduced its vehicular emissions in the last decades. However, it is still affected by air pollution events, mainly in the winter, characterized as a dry season. The chemical composition of fine particulate matter (PM$_{2.5}$) was studied in the MASP during a 100 d dry period in 2019. PM$_{2.5}$ samples underwent an extensive chemical characterization (including inorganic and organic species), ecotoxicity was assessed using a bioluminescence-based assay, and submicrometer particle number size distributions were simultaneously monitored. PM$_{2.5}$ concentrations exceeded the new World Health Organization's daily guidelines on 75 % TS1 of sampling days, emphasizing the need for strengthening local regulations. Source apportionment (positive matrix factorization, PMF5.0) was performed, and the sources related to vehicular emissions remain relevant (over 40 % of PM$_{2.5}$). A high contribution of biomass burning was observed, reaching 25 % of PM$_{2.5}$ mass and correlated with sample ecotoxicity. This input was associated with north and northwest winds, suggesting other emerging sources besides sugarcane burning (forest fires and sugarcane bagasse power plants). A mixed factor of vehicular emissions and road dust resuspension increased throughout the campaign was related to stronger winds, suggesting a significant resuspension. The sulfate secondary formation was related to humid conditions. Additionally, monitoring size particle distribution allowed the observation of particle growth on days impacted by secondary formation. The results pointed out that control measures of

high-PM$_{2.5}$ events should include the control of emerging biomass-burning sources in addition to stricter rules concerning vehicular emissions.

## 1  Introduction

According to the World Health Organization (WHO), a large part of the world's population lives in places where the recommended air quality standards are not achieved, and this includes the Metropolitan Area of São Paulo (MASP), Brazil (WHO, 2021; CETESB, 2023). In the MASP, thousands of metric tons of pollutants are released into the atmosphere every year. The MASP has more than 21 million inhabitants and around 7 million vehicles. In 2022, in the MASP, vehicles were responsible for 96 % of CO emissions, 70 % of HC, 60 % of NO$_x$, 8 % of SO$_2$, and 37 % of fine-mode particulate matter (PM$_{2.5}$) (CETESB, 2023). Among the pollutants, particulate matter is extensively studied, as it harms human health and is associated with cardiovascular diseases and cancer (Cohen et al., 2017). Furthermore, particulate matter has direct climatic effects through the absorption and scattering of solar radiation and indirect effects by affecting cloud microphysics (Li et al., 2022; Pöschl, 2005).

The fine fraction of particulate matter (PM$_{2.5}$) comprises particles with aerodynamic diameters equal to or below 2.5 µm that can penetrate the respiratory system, reaching the alveoli. Particles smaller than 100 nm are called ultrafine particles and can reach other organs through the lung vasculature (Schraufnagel, 2020; Kumar et al., 2014). Particles are emitted by different natural sources, such as volcanic eruptions, resuspension, and erosion of soils and vegetation, while the dominant anthropogenic sources are vehicles, industrial activities, and biomass burning. The formation of secondary particles is quite significant in some cities, such as São Paulo, where sulfate and nitrate concentrations are high and secondary particles can comprise half of PM$_{2.5}$ mass (Andrade et al., 2012; CETESB, 2023). Particulate matter is a complex mixture, including inorganic species (water-soluble ions and element oxides) and carbonaceous species, such as elemental carbon (EC, associated with soot) and organic carbon (OC) associated with organic compounds. Some particulate organic species present toxic properties, such as polycyclic aromatic hydrocarbons (PAHs) (Ravindra et al., 2008; Pöschl, 2005).

The vehicle fuel profile in the MASP is unique compared to other metropolises worldwide, with a significant proportion of biofuels, especially in light-duty vehicles (LDVs) (Andrade et al., 2017). The fleet has been running on gasoline and bioethanol blends (gasohol, 73 % of gasoline and 27 % of ethanol), hydrated ethanol (5 % of water), diesel and biodiesel blends (9 % of biodiesel produced from soybean) (Pereira et al., 2023a, b). By late 2018, ethanol usage peaked, reaching levels comparable to those of gasohol (Pereira et al., 2023a). Although the vehicle fleet has significantly increased since the 1980s, particulate pollutants have decreased over the past few decades (Andrade et al., 2017). Furthermore, a rise in the relative contribution of non-exhaust sources is expected in the next years due to control measures focused on vehicle exhaust (Thorpe and Harrison, 2008). A recent source apportionment study conducted in a densely populated area of the MASP, which experiences heavy vehicle traffic and has numerous industries, found that in 2019, light-duty vehicles contributed to 9.9 % of PM, while heavy-duty vehicles (HDVs) accounted for 42 % (Vieira et al., 2023). Sugarcane biomass burning typically influences air quality in the MASP during the winter (characterized as dry season); however, its occurrence in the north and northwest of São Paulo state also has been reduced due to control measures (Valente and Laurini, 2021). In addition to the significant levels of pollutants generated locally or regionally, increasing events of transport of biomass burning from the Amazon and central parts of Brazil were observed in the MASP at the end of the decade (Pereira et al., 2021; de Miranda et al., 2017), occurring mainly in the drier periods (July–October) (Vieira et al., 2023). Local biomass burning was pointed out as relevant, including the burning of charcoal and wood in barbecue and pizzerias and waste burning (Kumar et al., 2016). In 2014 and 2015, biomass-burning sources were apportioned and found to explain nearly one-fourth of particulate matter in the dry season (Pereira et al., 2017b; Emygdio et al., 2018).

The São Paulo State Environmental Company (CETESB) monitors legislated pollutants at 26 air quality stations throughout the MASP. Despite the efforts of local governments to improve air quality in the region, in 2019, they all recorded concentrations of PM$_{2.5}$ above the value recommended by the WHO, reaching critical levels in the drier months (April to September). Considering the aspects mentioned before, it is crucial to periodically update the chemical characterization of particulate matter since Brazilian environmental agencies do not carry out such extensive monitoring. It can be said that particle sources in São Paulo always need to be studied and identified, but this depends on good-quality chemical composition data. The composition of aerosols varies with the changing fleet and the policies concerning the burning of biomass. Furthermore, they are also influenced by the changing climate and meteorological conditions. In recent years, dryer weather conditions in the winter have favored forest and crop fires in central Brazil and the accumulation of pollutants (Souto-Oliveira et al., 2023; Pereira et al., 2021). This study aimed to thoroughly characterize PM$_{2.5}$ chemical composition and toxicity in the MASP,

identifying the relative contribution of emission sources. Associations with weather conditions and case studies of air pollution events were also investigated. The adoption of receptor models was performed to study the emission sources of PM$_{2.5}$, and these results were associated with particle size distributions.

## 2 Materials and methods

### 2.1 Sampling of PM$_{2.5}$ and particle number size distribution monitoring

High-volume samplers (Energética, Brazil) (1.13 m$^3$ min$^{-1}$ flow) were employed to collect PM$_{2.5}$ on the rooftop of the Institute of Chemistry building at the University of São Paulo, in the city of São Paulo, Brazil (23°33′53″ S, 46°43′32″ W), located in a green area and more than 1 km away from a busy expressway (Marginal Pinheiros) (Fig. 1). The sampling occurred between 4 June and 12 September TS2 2019, in a pre-lockdown polluted period, starting 6 months before the pandemic. During this campaign, a dark precipitation phenomenon (19 August) was observed – an unprecedented event that was termed "black rain" following biomass-burning episodes in central South America (Pereira et al., 2021). Ninety-nine samples were collected for 24 h (starting at 09:00 local time). Quartz fiber filters (20 cm × 25 cm, Whatman, UK) were employed to collect PM$_{2.5}$. These filters were decontaminated before sampling by heating at 600 °C for 6 h. Before and after sampling, the filters were weighed in a microbalance (controlled temperature and humidity for equilibrium: $T = 25$ °C and RH = 50 %). Then, the filters were wrapped into a laminated sheet and stored at 5 °C to avoid volatilization and reactions of the analytes before the analysis (de Oliveira Alves et al., 2015; Souza et al., 2014).

Particle number size distributions (PNSDs) were monitored using a scanning mobility particle sizer (SMPS 3081, TSI Inc.) in association with a condensation particle counter (CPC 3010, TSI Inc). The system provided particle number size distributions in the size range from 10 to 450 nm every 2 min. PNSD measurements were averaged to match the filter sampling periods. Average distributions were also calculated for periods of interest in the dataset. The SMPS data were collected on the rooftop of the Institute of Astronomy, Geophysics, and Atmospheric Sciences of the University of São Paulo, 500 m from the Institute of Chemistry. PNSD measurements were taken between June and September 2019, totaling 48 non-consecutive sampling days during the austral winter.

### 2.2 Analytical procedures

Distinct chemical composition analyses were performed. Thus, pieces of the filters were sent to each laboratory. At the Institute of Chemistry (University of São Paulo), water-soluble ions (WSI) and PAHs were determined. The extraction of WSI from filters (4 cm$^2$) was performed with 10 mL of deionized water (Milli-Q, Merck Millipore, USA) under sonication for 15 min. Then, extracts were filtered with syringe filters (Millex-GV; 0.22 μm; PVDF), and cations and anions were quantified in an ion chromatograph (modules 819, 830, 833, 818, and 820, Metrohm, Switzerland). Anions F$^-$, Cl$^-$, NO$_2^-$, Br$^-$, NO$_3^-$, PO$_4^{3-}$, SO$_4^{2-}$, C$_2$O$_4^{2-}$, HCO$_2^-$, and C$_4$H$_2$O$_4^{2-}$ and cations Ca$^{2+}$, Mg$^{2+}$, K$^+$, NH$_4^+$, and Na$^+$ were quantified. Pereira et al. (2023a) describe the adopted columns and eluents. Fluka (Switzerland) analyte standards were used. Recoveries fell between 80 %–120 % and were obtained by adding known concentrations of standards to blank filters, which are reported in Table S1 in the Supplement.

Punches of approximately 20 cm$^2$ from the filters underwent extraction using ultrasonic baths with 80 mL of dichloromethane for three cycles of 20 min, as described by Pereira et al. (2017a). The extracts were concentrated via rotary evaporation under low pressure (at 35 °C for approximately 30 min) and fractionated on a chromatographic column packed with 1.5 g silica gel. The methodology described in Vasconcellos et al. (2010) was modified and adopted in this study, involving three elutions: the first fraction carried alkanes (10 mL of hexane), while the second (9.6 mL of hexane + 5.4 mL of toluene) and the third (7.5 mL of hexane + 7.5 mL of dichloromethane) carried PAHs and their derivatives. The second and third fractions were combined to enhance recovery. The fractionated extracts were concentrated by rotary evaporation under reduced pressure (at 35 °C for approximately 15 min), filtered, stored in 2 mL vials, dried, and reconstituted with 0.5 μL of hexane. Samples were diluted prior to analysis by gas chromatography coupled with mass spectrometry (GC/MS, Agilent, GC 7820A and MS 5975), using an Agilent VF-5ms column (stationary phase, 30 m × 0.250 mm × 0.25 μm), with helium as the carrier gas (99.97 % purity and flow rate of 1.0 mL min$^{-1}$). The following species were determined: phenanthrene (Phe), anthracene (Ant), fluoranthene (Flt), pyrene (Pyr), retene (Ret), benzo(a)anthracene (BaA), chrysene (Chr) (low-molecular-weight PAHs; LMW-PAHs), benzo(b)fluoranthene (BbF), benzo(k)fluoranthene (BkF), benzo(e)pyrene (BeP), benzo(a)pyrene (BaP), indene(1,2,3-c,d)pyrene (InP), dibenzo(a,h)anthracene (DBA), benzo(g,h,i)pyrene (BPe), and coronene (Cor) (high-molecular-weight PAHs; HMW-PAHs). Recovery was assessed by adding known quantities of a PAH mix to filters containing 15 mg of the certified material (Urban Dust SRM 1649b, NIST, USA). Most species presented values between 80 % and 120 %, averaging 100 % (Table S1).

Carbonaceous species (OC and EC) in the quartz fiber filters were determined through thermal–optical analysis using a Sunset Laboratory Inc. carbon analyzer at the Institute of Physics of the University of São Paulo. The EUSAAR-2 tem-

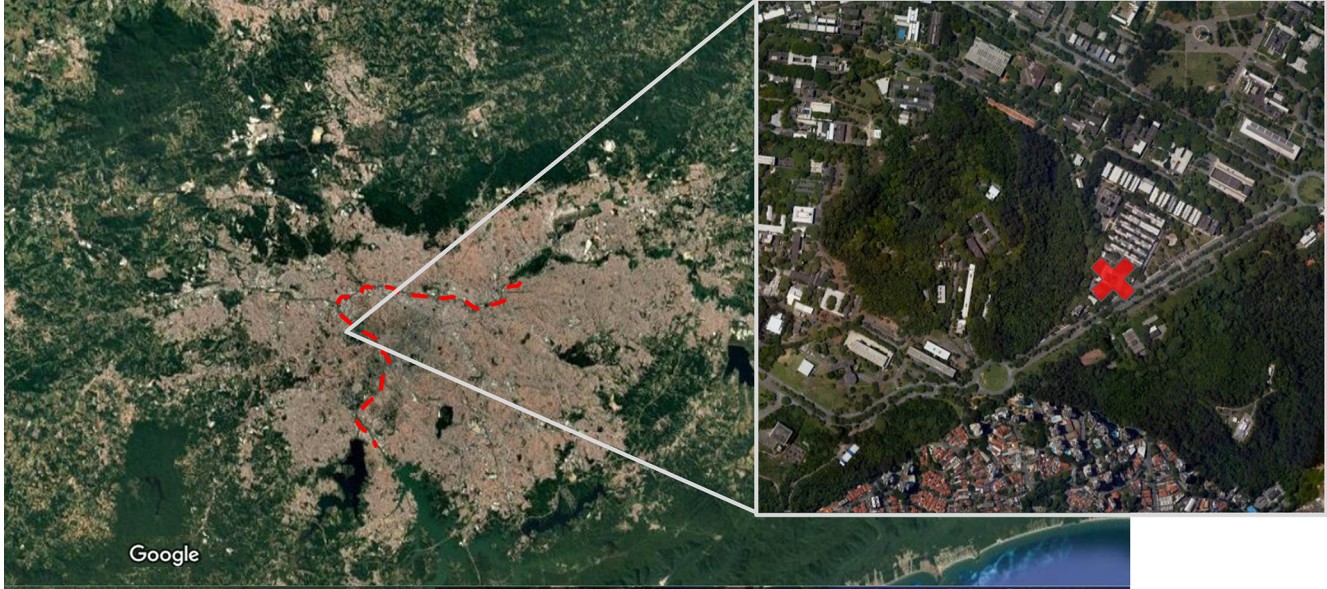

**Figure 1.** Sampling site location (marked with red cross) (© Google Maps). Heavily trafficked expressways are indicated with the dashed red line.

perature protocol (Cavalli et al., 2010) was employed, and the transmittance-based pyrolysis correction was applied, similar to previous studies conducted in the MASP (Monteiro dos Santos et al., 2016). Temperature-resolved carbon fractions were obtained, with increasing temperatures – four for organic fractions (OC1 to OC4, from higher- to lower-volatility temperatures) and four for elemental carbon (EC1 to EC4) – and the organic pyrolyzed carbon (PC) is monitored. Total OC is the sum of OC1, OC2, OC3, OC4, and PC, while total EC is the sum of EC1, EC2, EC3, and EC4, subtracted by PC. The first stage in helium medium is carried out under the following conditions of temperature and time steps: 200 °C for 120 s (OC1), 300 °C for 150 s (OC2), 450 °C for 180 s (OC3), and 650 °C for 180 s (OC4). In the second stage, under He–O$_2$ medium, the four steps are 500 °C for 120 s (EC1), 550 °C for 120 s (EC2), 700 °C for 70 s (EC3), and 850 °C for 80 s (EC4). Secondary organic carbon (SOC) was estimated considering the 5th TS3 percentile of the OC / EC ratios, similar to Monteiro dos Santos et al. (2016). Organic matter (OM) was calculated as 1.6 × OC, adopted for urban aerosols (Turpin and Lim, 2001).

Quartz fiber filters were subjected to acid digestion using a microwave digester oven (CEM MDS-2000, USA) at the Institute of Geosciences of the University of São Paulo. A strip of 20 cm × 2.5 cm of the filter was submitted to digestion in a closed vessel (PFA) with a mixture of HNO$_3$ and H$_2$O (10 and 15 mL, respectively), and then the volume was adjusted to 50 mL. Major and trace elements were quantified by inductively coupled plasma mass spectroscopy (ICP-MS, model iCAP Q, Thermo Fisher Scientific, USA). The percent

recoveries and detection limits were previously reported for all the determined analytes in Pereira et al. (2023a).

At the Centre for Environmental and Marine Studies (CE-SAM) of the University of Aveiro (Portugal), sugar species were extracted from the filters with ultrapure Milli-Q water in ultrasonic agitation, as described by Oduber et al. (2021). After the extraction, the solutions were filtered with syringe filters (0.2 µm; PTFE) and transferred to vials for liquid chromatography analysis with amperometric detection. Sugar compounds such as levoglucosan (Lev), mannosan (Man), galactosan (Gal), mannitol (Mnt), arabitol (Ara), and xylitol (Xyl) were determined with a Thermo Scientific Dionex™ ICS-5000 equipped with an anion-exchange analytical column (CarboPac® PA-1; 2 × 250 mm). Multi-step gradient conditions were adopted, with ultrapure Milli-Q water and two solutions of NaOH (200 and 5 mM). Recoveries were reported by Caseiro et al. (2007).

## 2.3 Benzo(a)pyrene equivalent indexes

Benzo(a)pyrene equivalent indexes were calculated to evaluate toxicity parameters due to exposure to PAHs. Equations (1) and (2) were employed to calculate the BaP$_{TEQ}$ and BaP$_{MEQ}$ by multiplying each species' concentrations by their toxic and mutagenic equivalency factors (TEF and MEF), as reported in de Oliveira Alves et al. (2020). The PAH carcinogenicity equivalent index (BaP$_{Eq}$) was also calculated by applying Eq. (3), incorporating specific PAH concentrations as adopted by Yassaa et al. (2001) and Cecinato (1997).

$$\begin{aligned}
\mathrm{BaP_{TEQ}} =\ & ([\mathrm{BaA}] \times 0.1) + ([\mathrm{Chr}] \times 0.01) \\
& + ([\mathrm{BbF}] \times 0.1) + ([\mathrm{BkF}] \times 0.1) + ([\mathrm{BaP}] \times 1) \\
& + ([\mathrm{InP}] \times 0.1) + ([\mathrm{DBA}] \times 5) \\
& + ([\mathrm{BPe}] \times 0.01)
\end{aligned} \tag{1}$$

$$\begin{aligned}
\mathrm{BaP_{MEQ}} =\ & ([\mathrm{BaA}] \times 0.082) + ([\mathrm{Chr}] \times 0.017) \\
& + ([\mathrm{BbF}] \times 0.25) + ([\mathrm{BkF}] \times 0.11) \\
& + ([\mathrm{BaP}] \times 1) + ([\mathrm{InP}] \times 0.31) \\
& + ([\mathrm{DBA}] \times 0.29) + ([\mathrm{BPe}] \times 0.19)
\end{aligned} \tag{2}$$

$$\begin{aligned}
\mathrm{BaP_{Eq}} =\ & ([\mathrm{BaA}] \times 0.06) + ([\mathrm{BbF}] \times 0.07) \\
& + ([\mathrm{BkF}] \times 0.07) + ([\mathrm{BaP}] \times 1) + ([\mathrm{DBA}] \times 0.6) \\
& + ([\mathrm{InP}] \times 0.08)
\end{aligned} \tag{3}$$

## 2.4 Data treatment and positive matrix factorization

Pearson coefficients were obtained to estimate the correlation between different variables (Jamovi software), and $r$ was considered significant when $p < 0.05$. Correlations between 0.0 and 0.3 were considered negligible, those between 0.3 and 0.5 weak, those between 0.5 and 0.7 moderate, those between 0.7 and 0.9 strong, and those between 0.9 and 1.0 very strong (Khan et al., 2018). To evaluate equal and unequal variances, the Mann–Whitney U test was also employed ($p < 0.05$). Polar plots were constructed with mass concentrations as wind speed and direction functions. They are obtained by a function in the openair (Carslaw and Ropkins, 2012) package (RStudio). Diagnostic ratios were performed between two (or more) chemical species concentrations: OC / EC, EC / Cu, Fe / Ca$^{2+}$, Cu / Sb, Cu / Zn, Fe / Cu, Sn / Sb, La / Ce, V / Ni, Pb / Cu, Cu / Ca$^{2+}$, BaP / (BaP + BeP), Flt / (Flt + Pir), InP / (InP + BPe), BaA / (BaA + Cri), LMW-PAHs / HMW-PAHs, Pyr / BaP, Pyr / BbF, Flt / BbF, NO$_3^-$ / EC, SO$_4^{2-}$ / NO$_3^-$, SO$_4^{2-}$ / EC, K$^+$ / Lev, Lev / Man, SO$_4^{2-}$ / Zn, and NO$_3^-$ / Zn.

The ISORROPIA model calculates the composition and phase state of the water-soluble inorganic aerosol in thermodynamic equilibrium with gas-phase precursors, simulating the process of dissolution of atmospheric particles and the ion formation (Bian et al., 2014; Fountoukis and Nenes, 2007; Li et al., 2014; Vieira-Filho et al., 2016a). In this study, ISORROPIA II was applied. It can solve two types of problems: (i) forward (or "closed system") and (ii) reverse (or "open system"). The reverse problem approach was adopted in this study using aerosol-phase concentrations of NH$_3$ (as ammonium ion), H$_2$SO$_4$, Na$^+$, Ca$^{2+}$, K$^+$, Mg$^{2+}$, HCl, and HNO$_3$ to estimate water content, salt aerosol concentrations, and gaseous precursors of aerosols. The aerosol–atmosphere system was considered thermodynamically stable (with precipitation of salts).

The enrichment factor (EF) is an approximation employed to identify the degree to which an element in the aerosol is enriched or depleted (da Rocha et al., 2012). EF was calculated using aluminum as a soil tracer, according to crustal element abundances described in the Appendix of Lee (1999). Elements with EF below 10 are of crustal origin (not enriched), and elements with EF above 10 are of non-crustal origin (anomalously enriched) (Pereira et al., 2007). Equation (4) is adopted to calculate EF, where $C_{\mathrm{Xp}}$ and $C_{\mathrm{Alp}}$ are the concentrations of elements X and Al in the sample, and $C_{\mathrm{Xc}}$ and $C_{\mathrm{Alc}}$ are their average concentrations in the Earth's crustal material:

$$\mathrm{EF} = \frac{\frac{C_{\mathrm{Xp}}}{C_{\mathrm{Alp}}}}{\frac{C_{\mathrm{Xc}}}{C_{\mathrm{Alc}}}}. \tag{4}$$

The positive matrix factorization (PMF) receptor model, including 94 samples, was applied to the datasets (Paatero and Tapper, 1994). The US EPA PMF 5.0 software was used. The variable classification followed the established definitions: "bad" when the signal-to-noise ratio ($S/N$) was below 0.2, "weak" when $S/N$ was between 0.2 and 2, and "strong" when $S/N$ was above 2 (Lang et al., 2015). The number of samples below the detection limit (Amato et al., 2016; Contini et al., 2016; Paatero and Hopke, 2003) and the thermal stability of the species (Pereira et al., 2017b) were also considered. Weak variables had their uncertainty increased by 3 times, and bad variables were excluded from the model (Norris et al., 2014). Twenty-two species were considered strong (NH$_4^+$, K$^+$, NO$_3^-$, SO$_4^{2-}$, V, Mn, Ni, Zn, As, Rb, Cd, Sb, Pb, Lev, OC1, OC2, OC3, OC4, PC, EC1, EC2, and EC3), six species were classified as weak (Ca$^{2+}$, Mg$^{2+}$, Cu, EC4, LMW-PAHs, and HMW-PAHs), and the concentrations of particulate matter were defined as a total variable (weak). Concentrations below the detection limits (DL) were substituted by half of the DL value. Uncertainties were calculated following the procedure by Norris et al. (2014). To evaluate the number of factors, $Q$ robust ($Q_{\mathrm{R}}$) was compared to $Q$ theoretical value ($Q_{\mathrm{T}}$) as in Pereira et al. (2017b). The final model was obtained with an additional 7 % modeling uncertainty. Solutions with three to eight factors were tested, and a final solution with five factors was taken as the best result, with factors that could be well interpreted. Bootstrap mapping (BS) and displacement of factor analysis (DISP) were used to analyze factor solutions, confirming a 5-factor solution as the most feasible. Most markers used to identify the sources were within the BS interquartile ranges (box), and mapping ranged from 87 %–100 %. Furthermore, no DISP swaps were observed.

## 2.5 Backward air mass trajectories

For specific pollution events, backward air mass trajectories spanning 48 h were generated using the HYSPLIT model (Draxler and Rolph, 2003) via the READY (Real-time Environmental Applications and Display System) platform provided by NOAA (National Oceanic and Atmospheric Admin-

istration). Trajectory frequencies were calculated to illustrate the origin of air masses reaching the MASP in selected polluted periods, starting at 09:00 LT. The trajectories were calculated at the heights of 500 and 3000 `TS4` m above ground level (a.g.l.), based on GDAS meteorological fields with 1° resolution.

## 2.6   Ecotoxicity assays

The ecotoxicity of PM$_{2.5}$ was screened by the kinetic version of the *Aliivibrio fischeri* bioluminescence-based assay (Kováts et al., 2021). This bioassay mimics the respiratory metabolism of biological systems resulting from exposure to particulate matter. This assay provides an easy-to-quantify endpoint to assess the presence of toxic substances. Inhibition of the bacteria's metabolism by toxic substances is demonstrated by the attenuation of its natural light emittance. Two 17 mm diameter filter sections were cut and ground in an agate mortar. The samples were then transferred to 4 mL vials, to which 2 mL of ultra-pure water was added. The suspensions were prepared with continuous agitation. The manufacturer's reconstitution solution was used to rehydrate the lyophilized bacteria (strain NRRL-B-11177, from Lange Co.), which were then stabilized for 35 min at 12 °C. For each sample, serial dilutions in 2 % NaCl were prepared in 96-well plates. After adding the bacterial suspensions to the samples, the bioluminescence intensity was continuously read for the first 30 s by a Luminoskan Ascent luminometer (Thermo Scientific). The bioluminescence was reread after 30 min of contact. The Ascent software (Aboatox Oy, Finland) was employed to calculate the EC50 (concentration that causes 50 % inhibition of bioluminescence compared to the control). Depending on their toxic units (TU = 100 / EC50 %), samples were cataloged as non-toxic (TU < 1), toxic (1 < TU < 10), very toxic (10 < TU < 100), or extremely toxic (TU > 100).

## 3   Results and discussions

### 3.1   PM$_{2.5}$ chemical composition and general trend

The concentrations of particulate matter presented a wide variation. PM$_{2.5}$ concentrations ranged from 7 to 47 µg m$^{-3}$, averaging 24 µg m$^{-3}$ (Fig. 2a) and exceeding the new World Health Organization's (WHO) daily recommendations on 75 % of sampling days (15 µg m$^{-3}$). However, no national or local limits were surpassed (CONAMA, 2018; WHO, 2021). The period was characterized by low precipitation, and the variation of meteorological conditions (temperature, relative humidity, pressure, and irradiance is described in the Supplement (Fig. S1). In terms of mass fractions, the most abundant elements observed in the 2019 intensive campaign were K > Al > Cu > Fe (Table 1). These species are linked to dust resuspension, vehicular sources, and biomass burning. Potassium (K) is attributed to soil resuspension and biomass burning, aluminum (Al) is related to soil resuspension, and iron

(Fe) and copper (Cu) are associated with vehicular sources in the MASP (Brito et al., 2013; Pereira et al., 2017b, 2023a, b). The biomass-burning tracer levoglucosan (Lev) increased in specific periods (Fig. 2b) and will be further discussed. As observed in the 2014 intensive campaign, secondarily formed ions were the most abundant: $NO_3^- > SO_4^{2-} > NH_4^+$ (Fig. 2c and d). However, the profile has changed slightly, showing a greater abundance of nitrate over sulfate in most of the period, unlike the intensive campaigns of 2008, 2010, 2013, and 2014 (Table S2), when sulfate was predominant in PM (Pereira et al., 2019, 2017a, b; Souza et al., 2014). The total mass of the most common element oxides was estimated ($Al_2O_3$, $SiO_2$, $TiO_2$, $MnO$, and $Fe_2O_3$). Since Si was not determined in this study, $SiO_2$ was estimated based on the Al content (as 3 times the concentrations of $Al_2O_3$) (Alves et al., 2018). The sum of these oxides can be used to estimate the proportion of crustal species in the PM$_{2.5}$ (Almeida et al., 2006) and accounted for 7 %.

A predominance of organic matter was observed in PM$_{2.5}$, accounting for a mass fraction of 47 %, on average, followed by $NO_3^-$ and EC (each accounting for 10 %) (Fig. 2c). OM and EC represented a significant fraction of PM$_{2.5}$ – over 50 % on average. Similarly, in the eastern part of São Paulo (2019–2020), organic matter (OM) corresponded to nearly 40 % of PM$_{2.5}$ (Vieira et al., 2023). Previous studies (2008 and 2014) pointed to higher proportions of EC in PM$_{2.5}$ (Table S2). The elemental carbon fraction is reducing worldwide (Chow et al., 2022; Yamagami et al., 2019) (more details are presented in the Supplement), and a reduction trend of vehicular emissions of this species was observed in the MASP (Pereira et al., 2023a, b). The emission limits following a control program policy were upgraded in the early 2010s, similar to Euro 5 and Euro 4, for heavy-duty vehicles (HDVs) and light-duty vehicles (LDVs), respectively (Pacheco et al., 2017) and more recently in 2022 (CETESB, 2022). The control policies are associated with reducing carbonaceous species, especially EC (Pereira et al., 2023a, b). Furthermore, adopting biofuels (e.g., ethanol and biodiesel blends) reduces these pollutants' emissions (de Abrantes et al., 2009). The relative reduction of sulfate and EC levels is observed worldwide, as reported in the Supplement. Following a similar trend to that observed in other Latin American metropolises (Gómez Peláez et al., 2020), the precursor SO$_2$ reduced relatively more than NO$_x$ in the MASP in the last decades (Andrade et al., 2017), and its concentrations were 8-fold lower in 2019 compared to 2000 (Fig. S2) (CETESB, 2020). The transport sector became the dominant SO$_2$ source in the MASP from the 1980s to the early 2000s after the reduction of industrial emissions following the adoption of electrical boilers (Kumar et al., 2016). Since the early 2010s, S10 diesel and S50 gasoline have been adopted to control vehicular emissions, although older trucks were allowed to use S500 diesel (CETESB, 2015). From 2024, a resolution will establish new national specifications for road-use diesel oils, discontinuing S500 fuels totally (MME, 2024). Due to these

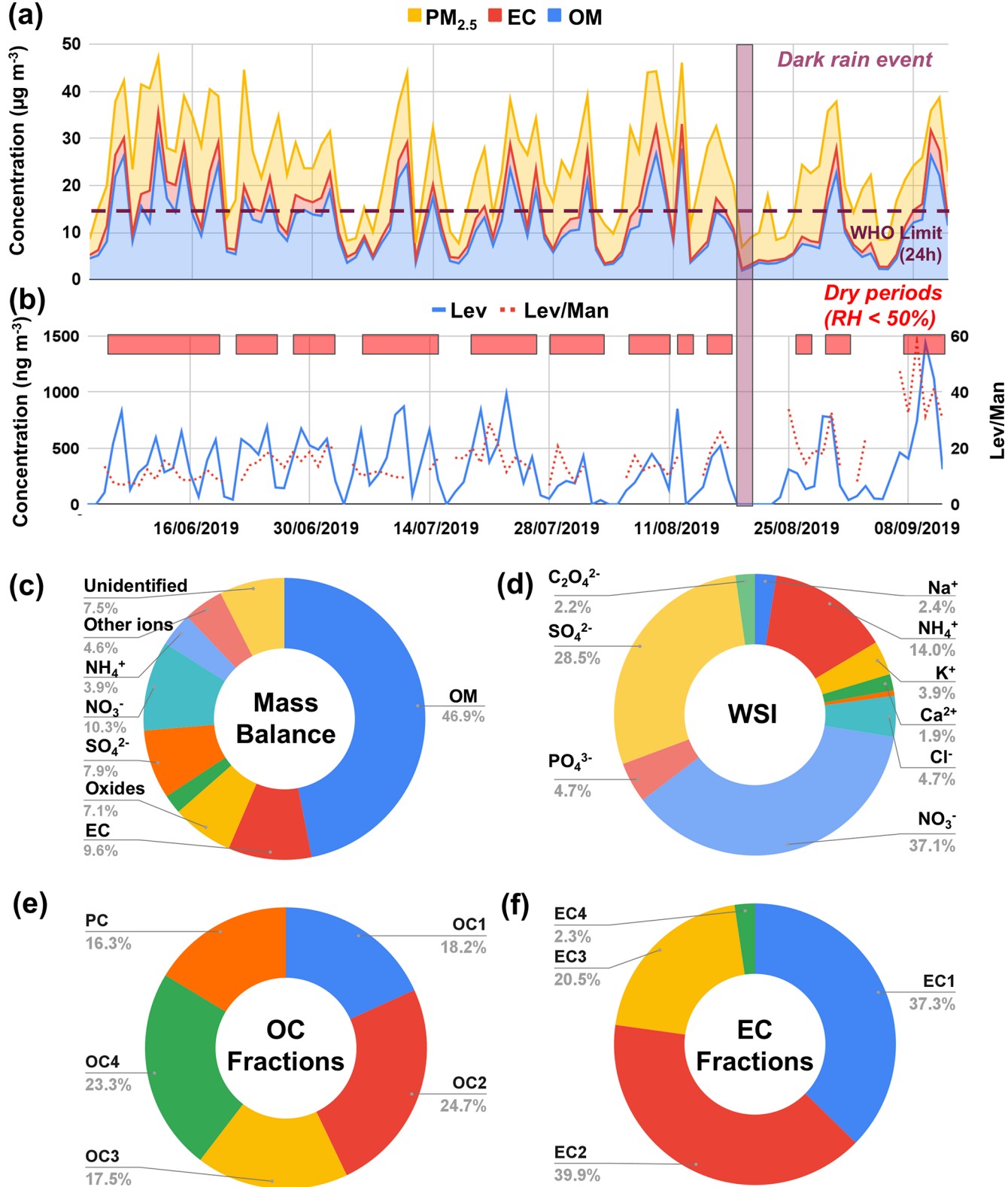

**Figure 2.** Daily variation in the concentrations of **(a)** PM$_{2.5}$ (the upper line gives concentrations, and the yellow area corresponds to non-carbonaceous PM$_{2.5}$), OM, and EC (stacked blue and red areas, respectively), and **(b)** Lev and Lev / Man ratio (the dark rain event is marked with the purple box, and dry periods when daily minimum RH < 50 % are marked with red squares). Pie charts for **(c)** mass balance, **(d)** water-soluble ionic composition, and **(e)** fractions of OC and **(f)** EC. CEI

**Table 1.** Median, average, minimum, and maximum concentrations of different species and diagnostic ratios*. TS5

| | Med. | Ave. | Min. | Max. |
|---|---|---|---|---|
| $(\mu g\,m^{-3})$ | | | | |
| $PM_{2.5}$ | 23.7 | 24.5 | 6.9 | 47.2 |
| OC | 6.3 | 7.2 | 1.2 | 18.6 |
| EC | 1.8 | 2.3 | 0.3 | 6.7 |
| SOC | 2.7 | 3.4 | 0 | 10.6 |
| $(ng\,m^{-3})$ | | | | |
| $Na^+$ | 143 | 166 | 5 | 549 |
| $NH_4^+$ | 737 | 954 | 61 | 4241 |
| $K^+$ | 247 | 268 | 6 | 686 |
| $Ca^{2+}$ | 101 | 128 | 17 | 517 |
| $Mg^{2+}$ | 44 | 43 | 6 | 95 |
| $Cl^-$ | 174 | 321 | 14 | 1505 |
| $NO_3^-$ | 2115 | 2531 | 134 | 10711 |
| $PO_4^{3-}$ | 320 | 318 | 36 | 784 |
| $SO_4^{2-}$ | 1651 | 1942 | 69 | 6139 |
| $C_2O_4^{2-}$ | 158 | 152 | 12 | 319 |
| $(ng\,m^{-3})$ | | | | |
| Lev | 315 | 380 | 38 | 1437 |
| Man | 23 | 29 | 7 | 123 |
| Gal | 13 | 16 | 7 | 41 |
| Xyl | 6 | 8 | 2 | 75 |
| $(ng\,m^{-3})$ | | | | |
| Al | 180 | 208 | 11 | 1142 |
| K | 271 | 286 | 16 | 757 |
| Ti | 7.5 | 8.9 | 1.4 | 141 |
| V | 1.2 | 1.7 | 0.1 | 8.0 |
| Cr | 1.7 | 3.4 | 0.0 | 20.7 |
| Mn | 6 | 6.9 | 0.6 | 25 |
| Fe | 64 | 99 | 2 | 674 |
| Co | 0.07 | 0.08 | 0.00 | 0.33 |
| Ni | 1.5 | 2.0 | 0.2 | 27 |
| Cu | 93 | 136 | 13 | 931 |
| Zn | 39 | 49 | 4.0 | 250 |
| As | 1.2 | 1.2 | 0.3 | 2.9 |
| Se | 1.1 | 1.4 | 0.1 | 5.7 |
| Rb | 1.8 | 2.2 | 0.1 | 6.2 |
| Sr | 4.6 | 5.2 | 0.3 | 15.4 |
| Mo | 1.6 | 4.9 | 0.1 | 31.2 |
| Ag | 0.27 | 0.26 | 0.01 | 0.85 |
| Cd | 0.49 | 0.66 | 0.03 | 2.69 |
| Sn | 5.5 | 8.2 | 0.2 | 30.7 |
| Sb | 3.8 | 5.7 | 0.3 | 27.9 |
| Ba | 14 | 16 | 4 | 54 |
| La | 0.25 | 0.29 | 0.01 | 1.47 |
| Ce | 0.23 | 0.26 | 0.01 | 2.09 |
| Tl | 0.06 | 0.07 | 0.00 | 0.23 |
| Pb | 11 | 14 | 1 | 38 |
| Bi | 0.16 | 0.2 | 0.01 | 0.9 |
| U | 0.23 | 0.25 | 0.06 | 0.69 |

| | Med. | Ave. | Min. | Max. |
|---|---|---|---|---|
| Ratios | | | | |
| OC / EC | – | 3.1 | 1.1 | 10.2 |
| EC / Cu | – | 17.3 | 2.2 | 146.5 |
| Fe / $Ca^{2+}$ | – | 0.8 | 0.0 | 7.8 |
| Cu / Sb | – | 23.6 | 3.6 | 543.1 |
| Cu / Zn | – | 2.8 | 0.2 | 23.1 |
| Fe / Cu | – | 0.7 | 0.0 | 6.8 |
| La / Ce | – | 1.1 | 0.5 | 16.7 |
| V / Ni | – | 0.8 | 0.1 | 2.8 |
| $SO_4^{2-}$ / $NO_3^-$ | – | 0.8 | 0.0 | 4.9 |
| $K^+$ / Lev | – | 0.7 | 0.0 | 4.7 |
| Lev / Man | – | 13 | 6.7 | 58 |

* Average diagnostic ratios were calculated with the average species' concentrations.

reductions in the MASP, sulfate vehicular emissions were estimated to be 5-fold lower than those from stationary sources in 2019 (CETESB, 2020).

The OC and EC fractions are shown in Fig. 2e and f, respectively. A predominance of OC2 and OC4 was observed among the organic carbon fractions and EC1 and EC2 among the elemental carbon fractions. The relatively low proportion of EC3 suggests that this site is less affected by HDV emissions since previous experiments in the MASP have shown that EC3 predominated in a diesel-powered HDV-impacted tunnel (Pereira et al., 2023b; Monteiro dos Santos et al., 2016). In the tunnel studies, the OC2 fraction dominated the emissions and had already been observed in urban areas of São Paulo in 2013 (Monteiro dos Santos et al., 2016). OC1 is the most volatile fraction, which may explain its relatively lower proportion in particulate matter. Pyrolyzed carbon (PC) represented 16 % of OC. Previously, PC represented 20 %–30 % of total OC at the university and downtown sites, accounting for less than 10 % of OC in a street canyon (Monteiro dos Santos et al., 2016). This fraction encompasses oxygenated components and links to water-soluble organic carbon (WSOC), secondary formation, and primary sources, including biomass burning (Zhu et al., 2014; Pio et al., 2007; Yu et al., 2002). More recently, it has been connected to HDV emissions, possibly due to the adoption of biodiesel (Pereira et al., 2023b).

Levoglucosan, a monosaccharide anhydride widely used as a biomass-burning tracer, presented the highest average concentration ($380\,ng\,m^{-3}$) among isomer sugars, followed by mannosan ($29\,ng\,m^{-3}$). Compared to a previous intensive campaign (2014), the average $K^+$ level reduced from 809 to $268\,ng\,m^{-3}$ (almost 3-fold), while levoglucosan presented a rather similar level ($509\,ng\,m^{-3}$ in 2014) (Pereira et al., 2017b). Since potassium levels are high in crop-burning emissions (Chow et al., 2022), the significant reduction in concentrations may be due to the lower influence of sugar-

cane burning. Control policies have contributed to the decrease in the number of fires (Valente and Laurini, 2021). Galactosan was present in less than half of the samples, averaging 16 ng m$^{-3}$. It is usually observed in lower proportions than mannosan in biomass-burning emissions (Bhattarai et al., 2019). It is possible to observe in the trajectory cluster a predominance of air masses coming from the north and northwest and a low influence of trajectories passing through the ocean (nearly 20 %) (Fig. 3a and b). This indicates the importance of the transport of biomass-burning aerosols from rural areas of São Paulo state and the central region of Brazil (nearly 50 %) during this period. The highest levoglucosan concentrations were observed on 10 and 11 September (1437 and 1119 ng m$^{-3}$) in a dry period that preceded a cold front. Relatively higher concentrations were also observed for K$^+$ on both days (543 and 555 ng m$^{-3}$). Back-trajectory frequencies arriving at 500 m for this 2 d period point to the typical influence of air masses from the north and northwest of São Paulo state (Fig. 3c and d), as observed in previous campaigns (Pereira et al., 2017a, b). When a height above the boundary layer is considered (3000 m), it is possible to see a frequency of trajectories passing through areas in the country's central region. The states of Minas Gerais and Goiás, located north of São Paulo state, also presented many fires in the studied period (Fig. S3) (INPE, 2019).

Xylitol was the most detected polyol, with an average concentration of 8 ng m$^{-3}$. This species is mainly found in biological material, soil biota, and biomass-burning smoke (Marynowski and Simoneit, 2022; Gonçalves et al., 2021; Caseiro et al., 2007). Gonçalves et al. (2021) found a relation between this polyol and biomass-fueled heating, reporting an increase from 2.95 ng m$^{-3}$ in summer to 19.9 ng m$^{-3}$ in winter in an urban background site in Portugal. However, much lower concentrations were observed in an urban site near sugarcane plantation areas in the north and northwest of São Paulo state (below 2 ng m$^{-3}$) (Carvalho et al., 2023). Back trajectories were obtained for a period with high xylitol levels, which peaked at 75 ng m$^{-3}$ on 30 August (Fig. S4). These trajectories indicate an influence of air masses from the north and northwest of São Paulo state, corroborating a biomass-burning contribution to this species (Fig. S5). Relatively high xylitol was also observed on 10 and 11 September (above 25 ng m$^{-3}$), when the highest levoglucosan concentrations were observed. Other polyols were detected in a few samples, as discussed in Supplement. Arabitol and mannitol levels tend to increase in the wet season, unlike biomass-burning tracers, as observed in a medium-sized city in the São Paulo state (Carvalho et al., 2023). The studied period is expected to present lower pollen-related emissions since these emissions are enhanced under higher temperatures and humidity (Marynowski and Simoneit, 2022).

The total concentration of PAHs in PM$_{2.5}$ ranged from 1.1 to 37.3 ng m$^{-3}$, with an average of 10.1 ng m$^{-3}$, which is higher than that observed in the 2019 extensive campaign (including a wet period), of 4.7 ng m$^{-3}$ (Serafeim et al., 2023).

The most abundant compounds observed in the present study were BbF, BeP, and BPe (Fig. 4 and Table S3). BbF is potentially carcinogenic and related to emissions from gasoline-powered vehicles (Ravindra et al., 2008), and its predominance is often observed in São Paulo (Pereira et al., 2017a, b). This compound is emitted in smaller amounts by LDVs than BaP (Pereira et al., 2023a). However, it is less influenced by chemical decomposition and is more persistent (Aubin and Farant, 2000). Along DBA, this species presents the longest photochemical residence time in the atmosphere among PAHs (Keyte et al., 2013).

Benzo(a)pyrene averaged 0.6 ng m$^{-3}$ (ranging from 0.04 to 2.39 ng m$^{-3}$) and surpassed the annual limit recommended by the European Environment Agency (EEA; 1 ng m$^{-3}$) in 16 % of the samples (Ravindra et al., 2008). However, half of the samples exceeded this value if the BaP toxicity equivalent index (BaP$_{TEQ}$) is considered (average of 1.9 ng m$^{-3}$ and range of 0.06–8.64 ng m$^{-3}$). The benzo(a)pyrene-equivalent index (BaP$_{Eq}$), calculated according to Yassaa et al. (2001), ranged between 0.04 and 4.08 ng m$^{-3}$ (average of 0.9 ng m$^{-3}$). BaP$_{Eq}$ was lower than that determined for samples collected in previous winter campaigns in São Paulo (ranging from 1.1 to 3.4 ng m$^{-3}$ in 2010, 2012, 2013, and 2014) (Pereira et al., 2017b, a). This reduction may be attributed to emission regulations and adoption of biofuels, such as ethanol and biodiesel, which can lower HMW-PAH emissions (de Abrantes et al., 2009; Pereira et al., 2023a). This trend follows a reduction in PAH emissions in recent decades in developed countries due to the establishment of new regulations (Shen et al., 2011). BaP$_{MEQ}$ and BaP$_{TEQ}$ tended to increase with northern and northwestern winds, which are more associated with drier weather, stable conditions, and biomass-burning-related aerosols (Fig. S6). The opposite was observed with S and SE winds when cold fronts and sea breezes were registered.

## 3.2 Diagnostic ratios and polar plots

The OC / EC ratios in the present study ranged from 1.1 to 10.2, with an average of 3.1 (the monthly averages varied from 3.0 to 3.3) (Fig. S7a). In previous studies in the MASP, ratios lower than 1 were often observed in tunnels and roadside experiments. These ratios increased to nearly 2 in greener, low-traffic sites, characterized by increased biogenic influence and secondary formation (Brito et al., 2013; Monteiro dos Santos et al., 2016; Pereira et al., 2023b). The distance to main roads can increase the ratios, and values between 1.8 and 3.7 are found in background sites (Amato et al., 2016). However, biomass burning can also increase this ratio, reaching over 4.1–14.5 (Watson et al., 2001; Wu et al., 2018). Notably, the monthly average values obtained in the present study were higher than those found in intensive campaigns in the same sampling site in 2014, 2013, 2010,

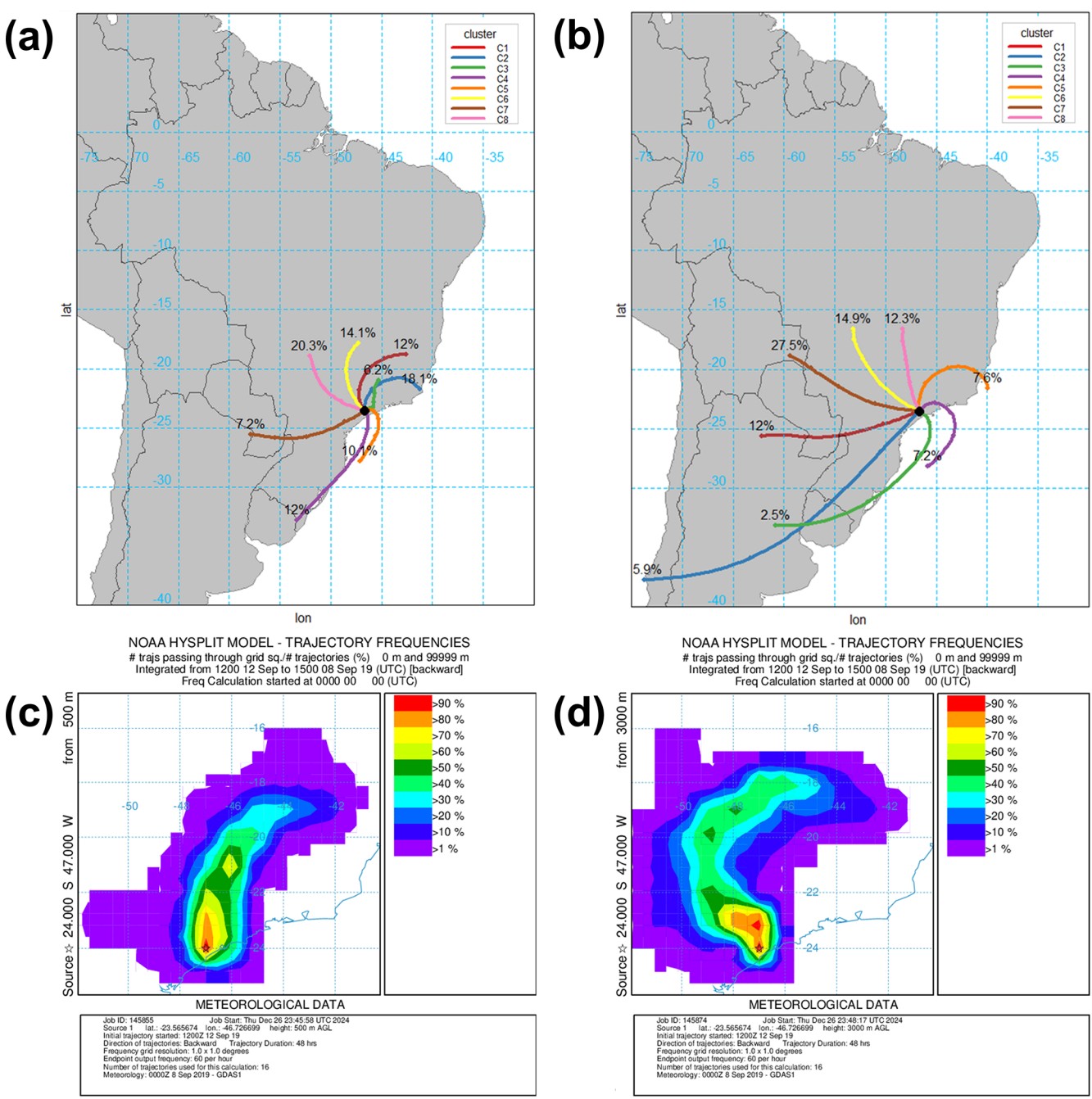

**Figure 3.** Backward trajectory clusters for the sampling period, arriving at **(a)** 500 m and **(b)** 3000 m, and trajectory frequencies for 10 and 11 September 2019, arriving at **(c)** 500 m and **(d)** 3000 m. TS6

and 2008 (1.2–2.3)[1]. These previous campaigns were conducted in similar dry periods but in shorter monitoring periods (Pereira et al., 2019, 2017a, b; Souza et al., 2014). The ratio increase between these years may indicate an improvement in engine efficiency alongside increased biofuel usage,

leading to a reduction of EC emissions (Dawidowski et al., 2024; Pereira et al., 2023a) since the ethanol sales in the state of SP in 2019 were 75 % higher than in 2013 (MME, 2023). Furthermore, an enhanced contribution from secondary organic carbon (SOC) formation may also contribute to this. SOC was estimated to reach nearly half of OC, with a maximum of 83 % (Table 1). The formation of SOC is enhanced under higher concentrations of oxidants such as ozone (Meng

---

[1]Ratios (OC / EC and Lev / Man) were obtained for PM$_{10}$ in 2010 and 2013, assuming that most of these species are found in the fine fraction in São Paulo.

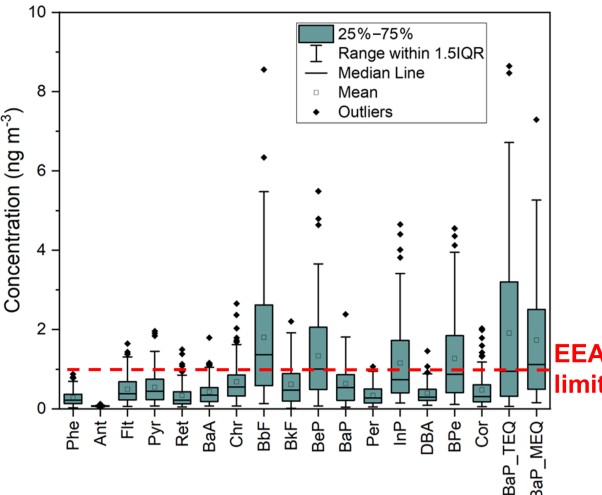

**Figure 4.** Box plots of polycyclic aromatic hydrocarbon concentrations. The dashed red line indicates EEA's recommended annual limit for BaP.

et al., 2020 TS7), which was observed in the MASP after 2006 (Andrade et al., 2017). In Santiago, Chile, there is a trend of increasing photochemical activity and oxidizing capacity in the atmosphere (2001–2018), leading to an increase in the secondary fraction of aerosols (Menares et al., 2020). The enhancement of SOC under a more oxidative atmosphere and/or higher temperatures requires further investigation.

Polar plots were obtained to investigate associations between wind direction and diagnostic ratios (Fig. 5). The OC / EC ratios appeared to increase with relatively stronger SE winds (Fig. 5a). These winds can also transport biogenic organic carbon from forested areas near the coast and air pollutants from coastal petrochemical and harbor areas. In the southeastern parts of MASP, which are more influenced by forests, less volatile oxidized organic aerosols were predominant and associated with SOC derived from volatile organic compounds (VOCs) emitted by biogenic and vehicular sources (Monteiro dos Santos et al., 2021). Furthermore, humidity can favor the partition of more polar SOC, as enhanced RH can lead to water uptake by hygroscopic submicron particles (Satsangi et al., 2021).

The average $SO_4^{2-}$ / $NO_3^-$ in this study was 0.8 (0.04–4.6 TS9), which is lower than the 1.6 (from July to December) of the 2019 extensive study (Serafeim et al., 2023). The difference can be attributed to the lower temperatures in the present study (from June to September), which are less favorable to the volatilization of particulate nitrate (Tang et al., 2016). In 2014, higher ratios were observed, reaching 1.2 in the intensive period and 2.2 in the extensive period (Pereira et al., 2017b). The ratio in the 2008 intensive campaign was higher than in 2014 (1.8) (Souza et al., 2014). Besides the temperature, the reduction in this ratio can be associated with the more accentuated reduction of sulfur dioxide levels if compared to nitrogen oxides (Sect. 3.1). The ra-

tio $SO_4^{2-}$ / $NO_3^-$ appeared to increase with SE and E winds (Fig. 5b). This suggests that the sulfate production may be favored by humid and cloudier conditions associated with these air masses. Additionally, this can be attributed to an influence of sulfur dioxide sources in the southeastern and eastern parts of the MASP (e.g., industries and HDVs). Sulfate particles can be formed by the growth of the condensation mode with the addition of sulfate and water in the aerosol droplet process (Guo et al., 2010). The southeastern part of MASP is influenced by industrial sources, and high sulfate levels are linked with locally emitted $SO_2$ (Monteiro dos Santos et al., 2021).

Galactosan and mannosan are products of hemicellulose thermal decomposition, while levoglucosan is formed during the combustion of cellulose (Simoneit, 2002). Since the amount of cellulose and hemicellulose varies with biomass type (hardwood contains relatively higher amounts of cellulose), the Lev / Man ratio can distinguish the smoke from different biofuels (Li et al., 2021; Zhu et al., 2015). Typical ranges of 15–25 and 3–10 have been documented for hardwood and softwood burning, respectively (Li et al., 2021). The average Lev / Man ratio in the present study was 13 (6.7–58.0 TS10), slightly higher than averages observed in 2014, 2013, and 2010 (ranging from 8 to 12)[1] TS11 (Pereira et al., 2017a, b, 2019). Previous ratios were closer to the values found in chamber sugarcane burnings (Lev / Man = 10) and areas impacted by this type of smoke plume (Lev / Man = 9) (Hall et al., 2012; Urban et al., 2014). In an agroindustrial region in northern São Paulo state, a change in the Lev / Man ratio was observed in 2020 (Lev / Man = 19) (Scaramboni, 2023). This increase was attributed to a change in the biomass-burning profile. Over the past decade, there has been a reduction in the burning of sugarcane straw and an increased use of sugarcane bagasse in power plants (MME, 2024). This shift, alongside the increased contribution of forest fires and other agricultural residue burnings, can play a significant role in this trend. During high-levoglucosan events in September, under lower relative humidity (Fig. 2), the Lev / Man ratios surpassed 40, suggesting the influence of different types of biomasses. In areas impacted by forest fires in the Amazon, likely linked to hardwood burning, the ratios ranged from 15 to 24 (Decesari et al., 2006; Graham, 2002). Furthermore, in central Brazil, burning grass in the *cerrado* (tropical savanna biome) resulted in relatively lower amounts of mannosan since this biomass contains less mannose than hardwood (Scaramboni et al., 2024). Before the dark precipitation event registered on 19 August, associated with smoke transported from areas in central Brazil and the Amazon (Pereira et al., 2021), Lev / Man values also exceeded 20.

The average $K^+$ / Lev ratio was 0.7, considerably lower than that observed in the 2014 dry period (1.6) (Pereira et al., 2017b). Lower $K^+$ / Lev ratios, typically below 1, have been found in emissions from woodstove combustion and forest fires (Caseiro et al., 2009). Jung et al. (2014) reported

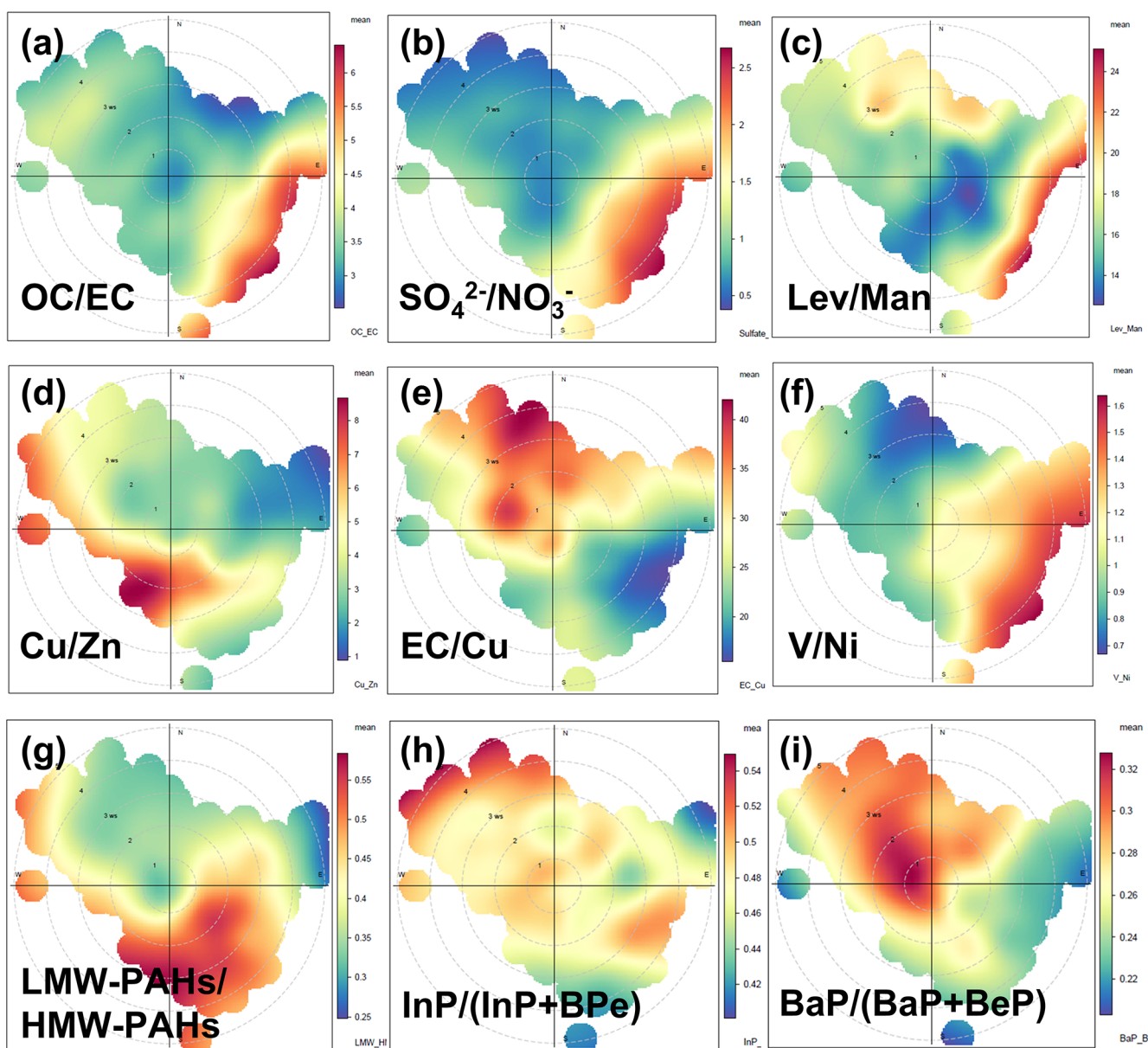

**Figure 5.** Polar plots considering the diagnostic ratios as a function of wind speed and direction for the ratios OC / EC (**a**), SO$_4^{2-}$ / NO$_3^-$ (**b**), Lev / Man (**c**), Cu / Zn (**d**), EC / Cu (**e**), V / Ni (**f**), LMW-PAHs / HMW-PAHs (**g**), InP / (InP + BPe) (**h**), and BaP / (BaP + BeP) (**i**). Distance from the center is related to wind speed, and ratios increase from blue to red. TS8

similar K$^+$ / Lev ratios for hardwood and softwood burning (0.04 and 0.02, respectively)[2] TS12, but this ratio was higher for crop burning (averaging 1.9)[2] TS13. Urban et al. (2012) documented an average ratio of around 4 for fine sugarcane-burning particles[2], justifying this value with the enrichment of K$^+$ in the leaves and the inefficient formation of levoglucosan during the flaming phase. The reduction in the proportion of K$^+$ / Lev in the present study may be associated with reducing sugarcane burning after regulation. Some K$^+$ / Lev

peaks were observed on days with very low Lev concentrations (Fig. S7b), suggesting the influence of soil-related K$^+$.

The Lev / Man ratio increased with stronger winds, whether from S/SE or N (Fig. 5c), denoting the influence of emissions from burning various types of biomasses. If the polar plot is obtained with conditional probability function (CPF) and non-parametric wind regression (NWR), the increase under stronger N winds is prominent (Fig. S8a and b). The lowest ratios were found with low-speed SE winds, probably due to wood burning in some restaurants in that area. Brazilian pizzerias traditionally use eucalypt logs in

---

[2]These authors presented the ratios as Lev / K$^+$.

woodstoves. However, recently, briquettes have been adopted (Lima, 2015). Sun et al. (2019) reported Lev / Man ratios in the range of 17–19 for different types of briquettes. Some authors found Lev / Man ratios mostly below 10 for woodstove emissions from different types of wood logs, while briquettes presented ratios of 29.7 in hot-start and 1.4 in cold-start conditions (Gonçalves et al., 2011).

The diagnostic ratios EC / Cu, Fe / Cu, Cu / Zn, and Cu / Sb were associated with the proportion of LDVs and HDVs in tunnels in the MASP (Pereira et al., 2023b). In previous studies, EC was more related to HDV emissions, while Cu was linked to LDV emissions (Brito et al., 2013). EC / Cu ratios in the present varied between 2 and 146, and the average ratio was 17. These values approached those observed for LDVs (5–8) rather than the ratios associated with a more HDV-impacted fleet (80–189). Sb has been described as a tracer of abrasion sources, including brake wear (Thorpe and Harrison, 2008). Cu is also present in Brazilian exhaust emissions from ethanol/gasohol vehicles (Brito et al., 2013). The present average Cu / Sb ratio was 24 (4–543), which is higher than the median values observed in tunnels (11–14). This difference may be attributed to the distance of the sampling site from a high-traffic area and a lower contribution from brake abrasion sources. The average Cu / Zn was 2.8 (0.2–23.1), falling between the median value observed for LDV (6.2) and that documented for HDV+LDV-impacted tunnels (0.9). The Cu / Zn and Cu / Sb ratios increased with southern weak winds (Figs. 5d and S8c), suggesting an enrichment in copper and predominance of LDV emissions in this area, possibly associated with the traffic of vehicles in the nearby residential neighborhood. On the other hand, the EC / Cu and Fe / Cu ratios increased with N winds (Figs. 5e and S8d), which may indicate an influence of HDVs passing in the expressway located north of the campus.

The average V / Ni ratio was 0.8 (0.1–2.8). Higher values for residual fuel oil combustion are found, ranging from 1 to 3 (Johnson et al., 2014). This ratio increased with strong E/NE/SE winds (Fig. 5f), suggesting an influence from oil burning in industries located in these areas (Vieira et al., 2023; Souto-Oliveira et al., 2021). The La / Ce ratio, similarly to the latter, was favored by S/SE winds (Fig. S8l). Higher ratios (4.3) were observed for fluidized-bed catalytic cracking (FCC) during petroleum refining, while a lower value (0.7) was documented for automobile catalyst emissions (Kulkarni et al., 2006). A petrochemical complex in the southeastern area of the MASP may explain this influence (Caumo et al., 2022; Gioia et al., 2017). Previous studies have identified isotopic fingerprints from the Cubatão petrochemical complex, an industrial area outside the MASP, following the passage of cold fronts and the arrival of southeastern winds (Souto-Oliveira et al., 2018).

Diagnostic ratios between PAHs for the analyzed samples were calculated and shown in Table S3. BaP / (BaP + BeP) values lower than 0.5 indicate that the analyzed particles were aged, as BaP undergoes photolysis more quickly than

BeP (Tobiszewski and Namieœnik, 2012). The average ratio was 0.32, suggesting a predominance of aged particles. **CE2** In three samples (collected on 27 July, 3 August, and 28 August), this ratio was close to 0.5, indicating fresh emissions. The Flt / (Flt + Pir) average was 0.48. According to De La Torre-Roche et al. (2009), values of this ratio ranging from 0.4 to 0.5 are characteristic of the burning of fossil fuels. InP / (InP + BPe) average ratio was equal to 0.48, similar to what was observed in 2014 by Pereira et al. (2017b). Values between 0.2 and 0.5 are associated with emissions from the burning of fossil fuels (Yunker et al., 2002). The BaA / (BaA + Cri) presented an average of 0.38, falling near that observed for vehicular sources (0.2–0.35) (Akyüz and Cabuk, 2010).

The ΣLMW-PAHs / ΣHMW-PAHs average ratio was 0.35, within the range reported for pyrogenic (< 1) emissions by some authors (Zhang et al., 2008). However, this ratio was previously found to vary with the proportions of LDVs and HDVs (0.7 and 7.5, respectively) (Pereira et al., 2023b) and temperature (Tobiszewski and Namieœnik, 2012). The ratios Pyr / BaP, Pyr / BbF, and Flt / BbF were associated with the proportion of LDVs and HDVs, increasing with the latter (Pereira et al., 2023b). Nevertheless, the ratios can be influenced by the volatilization of Pyr and Flt in warmer conditions and the photodegradation of BaP. The values were similar to those observed for gasoline emissions (near or below 1).

ΣLMW-PAHs / ΣHMW-PAHs and Pyr / BaP ratios appeared to be affected by strong winds coming from the NW (Figs. 5g and S8e), with mixed biomass-burning and vehicular influence. These ratios also increase under cooler winds from the S, which may favor the condensation of LMW-PAHs in the particulate phase (Ravindra et al., 2008). Temperature is observed as the meteorological parameter that most affects total and individual PAHs, with a more substantial influence on LMW-PAHs (Amarillo and Carreras, 2016). InP / (InP + BPe) ratios approached the values (above 0.5) found for grass, wood, and coal combustion with NW winds (Fig. 5h) (Yunker et al., 2002). It was not possible to observe the same for Flt / (Flt + Pyr) (Fig. S8f), which increased with S and SE winds. This suggests a relation between the long-range transport and the shorter photochemical residence time of Pyr compared to Flt (Keyte et al., 2013). The highest BaP / (BaP + BeP) values were registered for lower wind speeds (Fig. 5i), suggesting fresher local emissions (Tobiszewski and Namieœnik, 2012), although this ratio also seemed to increase with stronger NW winds.

## 3.3 Source apportionment (PMF)

The factor profiles obtained by positive matrix factorization are shown in Fig. 6. A five-factor solution was obtained, attributed to biomass burning (BB), secondary formation (SF), and industries (IN), and two were associated with vehicular emissions (VE1 and VE2). Factor 1 (BB) was charac-

terized by biomass-burning-related species such as levoglucosan, K$^+$, and some carbonaceous species (Bhattarai et al., 2019; Simoneit, 2002) (Fig. 6a). However, the presence of Cd, Sb, and Pb suggests a mixture with vehicular sources (Thorpe and Harrison, 2008) or that these species may be linked with waste burning (La Colla et al., 2021). This factor was also characterized by a significant contribution from pyrolyzed carbon (PC), which is linked to water-soluble organic carbon (WSOC), often associated with biomass-burning and secondary organic aerosol (Yu et al., 2002; Zhu et al., 2014). Recently, it has been observed that the biomass-burning factor can mix with secondary organic carbon contribution in a Korean industrial area (Han et al., 2023). Among elemental carbon fractions, EC1 was the most abundant. Some elements, such as Rb, also appeared in the biomass-burning profile. This species has been described as a component of some types of soil by Calvo et al. (2013), wood-burning emissions (Fine et al., 2001), and biomass-burning aerosols in the Amazon (Artaxo et al., 1994). Recently, water-soluble rubidium in fine particulate matter has been assigned to wood-burning emissions and considered an alternative biomass-burning tracer (Massimi et al., 2020). This factor increased with winds coming from the N and NW, indicating the influence of the transport of biomass-burning aerosols from agroindustrial areas in São Paulo state and potentially from forested areas in northern and central Brazil. It decreases with S and SW winds, suggesting a reduction with cold fronts and sea breezes (Sánchez-Ccoyllo and Andrade, 2002). The results agree with the study of Souto-Oliveira et al. (2023), which showed the impact of the transport of wildfires to the MASP during fine-particulate-matter exceedance events in 2020.

Factor 2 (SF) was highly loaded with secondary formation species, mainly NH$_4^+$ and SO$_4^{2-}$ (Fig. 6b). This process likely involves the oxidation of sulfur dioxide, the formation of sulfuric acid, neutralization with ammonia, and the production of ammonium sulfate ((NH$_4$)$_2$SO$_4$) (Han et al., 2023; Iannielo et al., 2011). Furthermore, this factor also presented loadings of some primary emission species, such as vanadium. Serafeim et al. (2023) also observed high loadings for Ni and V in the secondary related factor in the MASP. These species originate from industrial emissions (Calvo et al., 2013). Previously, it has been observed with the WRF-Chem model that secondary formation explains 20 % to 30 % of PM$_{2.5}$ in the MASP (Vara-Vela et al., 2016), increasing in the summertime (Pereira et al., 2017b). The SF factor was previously attributed to vehicular-related emissions in a previous study in 2014 (Pereira et al., 2017b). However, with the reduction of SO$_2$ emissions by vehicular sources (CETESB, 2020), the industries' contribution has become relevant (Sect. 3.1). However, most industries are located on the outskirts, whereas vehicular emissions are more concentrated in the city's central areas. Thus, the apportionment of secondary sulfate formation can be a subject for further studies.

Factor 3 (IN) showed high V, Ni, and nitrate loads (Fig. 6c). Earlier studies in the MASP have associated Ni and V with residual oil burning and industrial sources (Andrade et al., 1994; Castanho and Artaxo, 2001). Heavy oil combustion in industrial boilers also emits nickel and zinc, especially in the ultrafine fraction (smaller than 0.1 µm) (Jang et al., 2007; Linak et al., 2004). Factor 3 (IN) increased with the same wind direction as factor 2 (SF), which suggests that it may be partially related to secondary formation. This is possibly due to emissions of precursor gases, such as NO$_x$ and SO$_2$, or associated with the same air masses. Secondary aerosol formation can mislead the separation of factors in PMF, as observed by Faisal et al. (2024).

Factor 4 (VE1) was loaded with soil resuspension constituents, such as Ca$^{2+}$ and Mg$^{2+}$, as well as with species previously related to vehicular emissions in the MASP, such as As and Cu, carbonaceous species (OC1, OC2, OC3, OC4, EC3, and EC4), LMW-PAHs, and HMW-PAHs (Pereira et al., 2017b, 2023a, b) (Fig. 6d). In the extensive 2019 campaign, a single road dust factor represented 32 % of PM$_{2.5}$ (Serafeim et al., 2023). The association with construction-related calcium, found in concrete, can also be considered (Bourotte et al., 2006). EC3 and EC4 were abundant in this factor. EC3 was anteriorly related to HDV emissions in MASP, while EC4 was found in an urban canyon. Furthermore, OC2, OC3, and OC4 were found in LDV emissions in similar proportions (Monteiro dos Santos et al., 2016; Pereira et al., 2023a). Levoglucosan was also observed in a smaller proportion in this factor. As observed in 2014, this factor overlaps with the BB factor (Pereira et al., 2017b), as they increase with the same wind direction. However, the contribution of VE1 increases with NW stronger winds, corroborating effect of road dust resuspension. Strong NW winds are often attributed to prefrontal conditions (Ribeiro et al., 2018). Another aspect contributing to the overlap is that this anhydrosugar is also found in the PM$_{10}$ from the wear between tires and pavements, given that wheel rubbers have cellulose in their composition (Alves et al., 2020). However, PM$_{2.5}$-bound levoglucosan is emitted in much smaller proportions by this non-exhaust vehicle emission source compared to biomass burning (Bhattarai et al., 2019). This hybrid factor may occur because the aerosols in the urban atmosphere, comprised mainly of vehicular-related species, road dust, and biomass-burning smoke, arrive mixed at this semi-background site. Additionally, it is challenging to differentiate HDV from LDV emissions in this site (Pereira et al., 2017b). The increase in time resolution for PMF can reduce mixing in source profiles, decreasing sample variability (Wang et al., 2018). This improvement can be achieved through real-time techniques (e.g., aerosol chemical speciation monitor) (Monteiro dos Santos et al., 2021). The contribution of non-exhaust emissions, such as road dust resuspension, is expected to increase in the future (Thorpe and Harrison, 2008) as exhaust emissions are on a downward trend

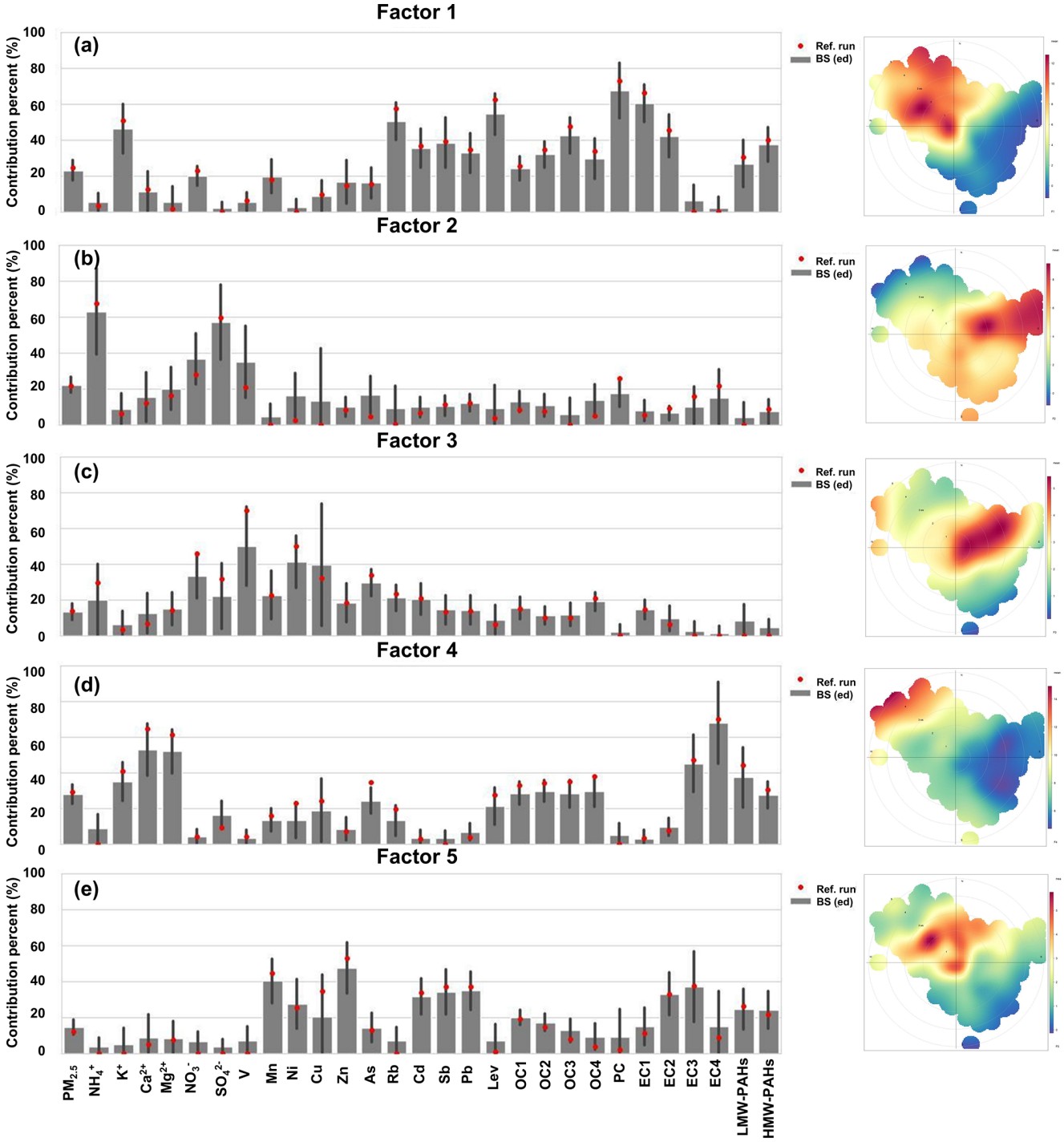

**Figure 6.** Profiles identified with the PMF receptor model and polar plots. **(a)** Factor 1: biomass burning (BB). **(b)** Factor 2: secondary formation (SF). **(c)** Factor 3: industrial (IN). **(d)** Factor 4: vehicular + road dust (VE1). **(e)** Factor 5: local vehicular (VE2). Model reference runs (Ref. run) are marked in red, and bootstrap runs (BS) and standard deviation (SD) are marked in gray.

due to increasingly stricter regulations and improved treatment systems.

Factor 5 (VE2) presented several species of vehicular origin, such as Mn, Ni, Cu, Zn, Sb, and the potentially toxic species Cd and Pb (Brito et al., 2013; Pereira et al., 2023a)

(Fig. 6e). OC and EC fractions were also found in this factor, particularly EC3, which is associated with HDVs (Pereira et al., 2023b). This vehicular factor increased with lower wind speed, suggesting a local contribution. The site is located next to an avenue with a constant flow of buses and LDVs

during the daytime and is near a busy expressway. Despite the lower proportion, the emissions of buses and trucks contributed to almost half of the black carbon levels observed in the MASP (Brito et al., 2018). In the scanning electron microscopy (SEM) study performed by Bourotte et al. (2006), particles rich in carbon, copper, and zinc were identified and associated with incomplete fossil fuel burning. Zinc is also abundant in tire wear particles, as zinc oxide is added to the tires in the vulcanization process (Thorpe and Harrison, 2008). Copper was previously assigned to LDV exhaust emissions and to the corrosion of the internal parts of the MASP fleet engines (Ferreira da Silva et al., 2010). This species was also pointed out as a brake dust particle tracer (Thorpe and Harrison, 2008).

Figure 7 shows the contributions of each type of source to PM$_{2.5}$ and their variation between sampling days. Biomass burning accounted for 25 % of PM$_{2.5}$. This contribution was slightly higher than that documented in the intensive campaign of 2014 (18.3 %). In the extensive study performed in 2019 (including dry and wet seasons), the biomass-burning factor represented 13 % of ambient PM$_{2.5}$, a similar share to that observed in the 2014 extensive period (Pereira et al., 2017b; Serafeim et al., 2023). However, it is lower than that observed in the present study since the period from June to September is typically more affected by biomass burning (Pereira et al., 2017a). In a major city in central Brazil (Cuiabá), biomass burning was relevant and represented one-third of PM$_{10}$ (Parra et al., 2024). Several peaks of the BB factor coincided with lower RH values. An increase in the contribution of this factor was observed 2 d before the sky-darkening phenomenon at MASP on 19 August, when the occurrence of dark precipitation was attributed to pollutants emitted by biomass burning (Pereira et al., 2021). Overall, crop burning and wildfires can explain a significant share of PM$_{2.5}$ in this period. However, the regional biomass-burning profile is changing, with reduced sugarcane straw burning, increased contribution of wildfires, and sugarcane bagasse used as fuel, as discussed in Sect. 3.2. The study of Knorr et al. (2017) predicts that due to climate change, levels of wildfire-related PM$_{2.5}$ can rise to dangerous levels during fire seasons in many regions, including parts of South America. This increase in PM$_{2.5}$ levels can occur even with accentuated reductions of other anthropogenic emissions. In the last decade, wildfire smoke has influenced PM$_{2.5}$ trends across much of the US, including major cities, complicating efforts to control this pollutant (Buchholz et al., 2022). In temperate regions, the high contribution of biomass burning to PM$_{2.5}$ in the winter can be attributed to residential heating (Nava et al., 2020). In the temperate metropolis of Beijing (China), biomass and coal burning can represent nearly half of PM$_{2.5}$ in winter, as these sources are used for heating and cooking in nearby rural areas (Srivastava et al., 2021). Due to regulations, open-field burning in China has become less frequent. However, biomass burning for home heating in north-ern China can affect downwind areas such as South Korea (Cheong et al., 2024).

The vehicular-related sources, VE1 (29 %) and VE2 (12 %), represent a significant share of particulate matter in the city (over 40 %). Vehicular sources were dominant in 2013, 2014, and 2015 (Souto-Oliveira et al., 2021; Emygdio et al., 2018; Pereira et al., 2017a, b). Previous studies were performed in other Brazilian cities, where vehicular emissions ranged from 3 % to nearly half of PM$_{2.5}$ (Rio de Janeiro, Vitória, and Recife). Sea spray was found relevant in Brazilian coastal sites (Galvão et al., 2019; Justo et al., 2023; dos Santos et al., 2014). Despite the entrance of sea breezes on a few occasions, it was not possible to identify this source in the present study. As discussed in Sect. 3.1, a few trajectories passed by the ocean. The factor VE1 increased throughout the sampling period and dominated PM$_{2.5}$ at the end of August and the beginning of September, indicating greater dust resuspension at the end of winter due to the long dry spell and higher wind speeds (Fig. S9). In another source apportionment study performed in the east region of São Paulo in 2019, four factors were found to contribute to PM$_{2.5}$: heavy-duty vehicles (42 %), light-duty vehicles (9.9 %), soil and local particles (38.7 %), and local sources (8.6 %) (Vieira et al., 2023), with a higher share of vehicular sources than the present study and an increase of soil particles in the dry season. According to Serafeim et al. (2023), vehicular emissions and biomass burning were associated with enhanced oxidative potential.

Secondary formation (SF) of aerosol from oxide precursors, which can be emitted from vehicular and industrial sources, represented 21 % of PM$_{2.5}$. Industrial sources accounted for only 13.6 %, a similar level to that observed in 2014 (over 10 %) (Pereira et al., 2017b). A relatively low impact of direct industrial emissions is felt in this part of the city. Most industries are now located on the MASP's outskirts and in other metropolitan areas in the state (Kumar et al., 2016), and there are no large industries near the sampling site. Industrial sources can be relevant in other Brazilian cities. Pelletizing and steel industry contributed to more than 60 % of PM$_{2.5}$ in Vitória (Galvão et al., 2019). The simplified quantitative transport bias analysis (SQTBA) approach was utilized, recognizing the plume dispersion process and considering it as part of back-trajectory analysis. The contribution of the IN factor, combined with the frequencies of the trajectories, suggested an influence from areas located to the east and northeast (Fig. 8a), including parts of the MASP and other neighboring metropolitan areas with a high concentration of industries (Fig. 8b).

## 3.4 Correlations between PMF results and other variables

The factors obtained with the PMF receptor model were correlated with other variables: meteorological data (temperature, relative humidity, and wind) collected from the

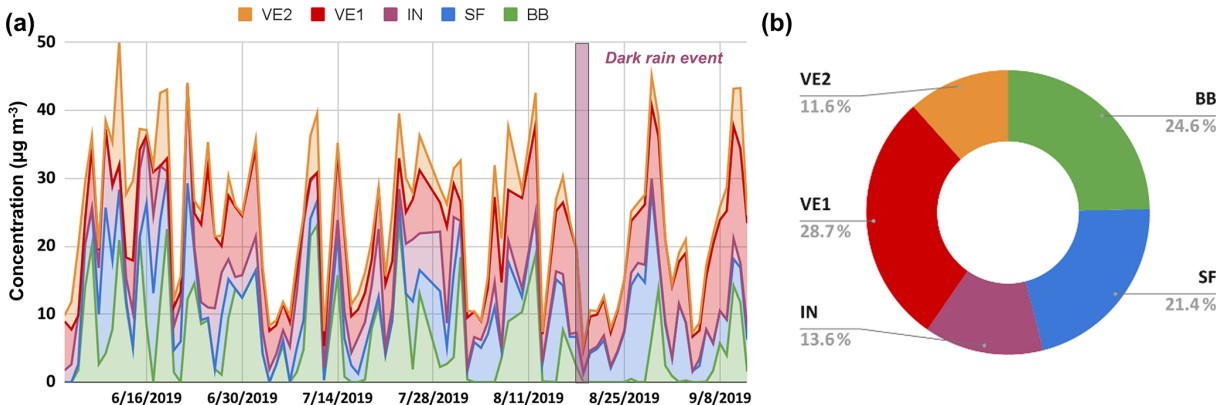

**Figure 7.** Daily factor contribution (**a**) and PM$_{2.5}$ contributions (**b**) identified with the PMF receptor model.

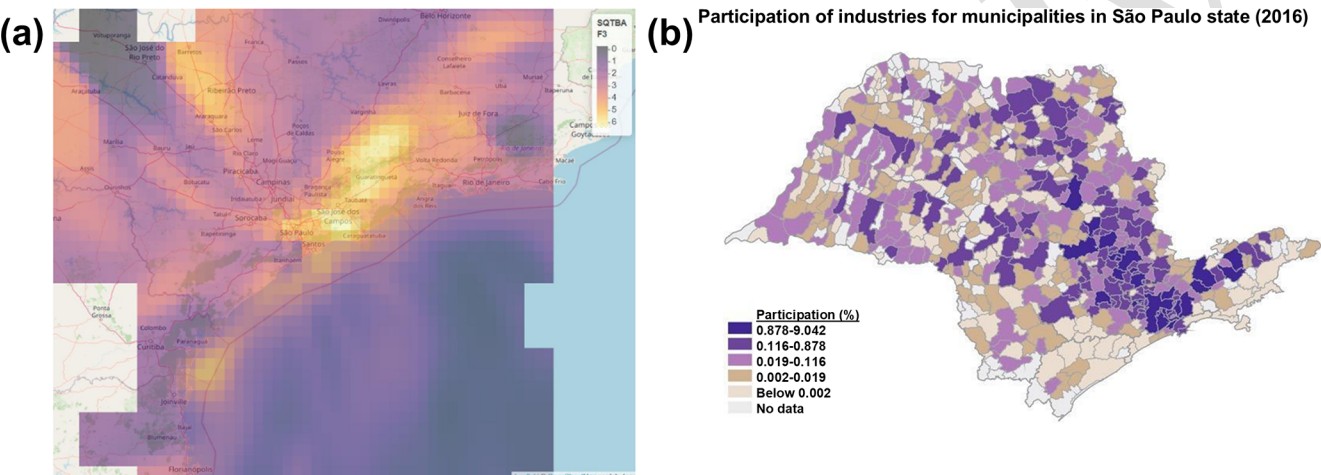

**Figure 8.** (**a**) Trajectory analysis for the contribution of factor 3 and (**b**) participation of industries for municipalities in São Paulo state (industrial transformation value), adapted from SEADE (2019).

local station, concentrations of chemical species, diagnostic ratios, and results obtained from the ISORROPIA thermodynamic model (modeled solid and liquid inorganic aerosol, water content, NaNO$_{3(s)}$, Na$_2$SO$_{4(s)}$, NaHSO$_{4(s)}$, NaCl$_{(s)}$, NH$_4$Cl$_{(s)}$, NH$_4$NO$_{3(s)}$, (NH$_4$)$_2$SO$_{4(s)}$, NH$_4$HSO$_{4(s)}$, CaSO$_{4(s)}$, Ca(NO$_3$)$_{2(s)}$, CaCl$_{2(s)}$, K$_2$SO$_{4(s)}$, KHSO$_{4(s)}$, KNO$_{3(s)}$, KCl$_{(s)}$, MgSO$_{4(s)}$, Mg(NO$_3$)$_{2(s)}$, MgCl$_{2(s)}$, H$^+_{(aq)}$, Na$^+_{(aq)}$, NH$^+_{4\ (aq)}$, Cl$^-_{(aq)}$, NO$^-_{3\ (aq)}$, SO$^{2-}_{4\ (aq)}$, HSO$^-_{4\ (aq)}$, Ca$^{2+}_{(aq)}$, K$^+_{(aq)}$, and Mg$^{2+}_{(aq)}$) (Table S4 and Fig. S10). Overall, the components were negatively correlated with RH and positively correlated with temperature, indicating dry air masses and unfavorable conditions for the dispersion of atmospheric pollutants (Sánchez-Ccoyllo et al., 2002). More discussions are presented in the Supplement.

The biomass-burning (BB) factor correlated with BPe and InP, species typically associated with vehicular emissions TS14 (Ravindra et al., 2008). However, the InP / (BPe+InP) polar plots suggested an influence of biomass burning with northwest strong winds (Sect. 3.2).

The low correlation of retene with BB and levoglucosan suggests that this species may be related to a local biomass-burning source or the influence of gas–particle partition of this semi-volatile species (Ravindra et al., 2008). Chloride was moderately correlated with BB and VE2 factors ($r > 0.5$), strongly correlated with mannosan and EC1 ($r > 0.7$), and weakly correlated with levoglucosan. This suggests a different biomass-burning profile, such as wood burning in restaurants or biomass burning associated with waste (Kumar et al., 2015). Xylitol was weakly correlated with BB and VE1 ($r > 0.3$). It also presented a moderate negative correlation with RH ($r \sim -0.3$), suggesting an increase under drier conditions. Notably, it presented weaker correlations with potassium compared to Lev, suggesting a biomass-burning origin that is less associated with crop burning. The association of xylitol with biomass burning was observed in other studies (Clemente et al., 2024; Gonçalves et al., 2021).

The secondary aerosol formation (SF) factor was moderately to strongly correlated with the modeled liquid in-

organic aerosol and water content of the aerosol, as well as the modeled secondary species $NH_{4\ (aq)}^+$, $SO_{4\ (aq)}^{2-}$, and $HSO_{4\ (aq)}^-$ ($r > 0.6$). Correlation with modeled water content suggests that the secondary formation pathway in São Paulo depends on humidity. $SO_4^{2-}$ and $NO_3^-$ formation in clouds and fog droplets were attributed mainly to the heterogeneous aqueous transformation of $SO_2$ and $NO_x$ under high relative humidity (RH) during haze episodes in Beijing (Huang et al., 2016). Sulfate particles can grow in the aerosol droplet process (Guo et al., 2010). Among the ratios, the SF factor presented weak to moderate correlations with $NO_3^- / EC$, $SO_4^{2-} / EC$, and $NO_3^- / Zn$ ($r > 0.4$), suggesting that they can help identify secondary formation events – an opposite trend to that observed for the BB, VE1, and VE2 factors.

The industrial (IN) factor presented moderate correlations with the liquid inorganic aerosol and water content in the aerosol, possibly associated with more humid air masses from the east observed in the polar plot, as particles can grow in the aerosol droplet process (Guo et al., 2010). Notably, it was correlated with aerosol acidity ($H_{(aq)}^+$), perhaps due to simultaneous emission of gaseous oxides that can undergo secondary reactions and produce acidic species. $H_{(aq)}^+$ was also moderately correlated with nitrate, suggesting an influence of nitric acid (Ianniello et al., 2011). Selenium, previously attributed to industrial sources in the MASP (Vieira et al., 2023), was moderately associated with the IN factor. Sn was moderately correlated with IN ($r \sim 0.5$) and strongly correlated with Tl. The IN factor presented a weak correlation with the V / Ni ratio ($r \sim 0.5$), denoting the contribution from the burning of crude oil (Johnson et al., 2014). IN was also moderately correlated with $NO_3^- / EC$, suggesting a connection with secondary formation. There was a weak correlation between SF and IN, pointing to a common origin between both, maybe due to the emission of gaseous precursors and secondary formation.

The PMF factor associated with vehicular exhaust and road dust resuspension (VE1) presented weak correlations with most species. The factor presented moderate correlations with the modeled mass of solid inorganic aerosol ($r \sim 0.6$). This factor also displayed a moderate positive correlation with temperature ($r > 0.6$) and a negative correlation with relative humidity and pressure ($r < -0.5$), suggesting an increase of this factor during prefrontal conditions.

The local vehicular source (VE2) presented higher correlations with BaA and HMW-PAHs such as Per, InP, BPe, and Cor ($r > 0.5$), which are found in LDV emissions (Pereira et al., 2023a). Among the identified factors, VE2 and BB presented higher correlations with the toxicity equivalent indexes BaP-TEQ and BaP-MEQ ($r > 0.5$), suggesting a contribution of these sources to potential carcinogenicity and mutagenicity. La was moderately associated with VE2 and strongly with Mn, Co, and Ce. La and Ce derive mainly from catalyst emissions (Kulkarni et al., 2006). Fe presented moderate to strong correlations with VE2, Ti, Mn, and Ce.

These species are linked to vehicular and soil-related sources (Pereira et al., 2017a; Brito et al., 2013). Cobalt, a component of tire debris (Thorpe and Harrison, 2008), was moderately correlated with VE2 and strongly correlated with La, Tl, and Mn. VE2 presented moderate correlations with Fe / Ca$^{2+}$ ($r \sim 0.6$), which may indicate enrichment of anthropogenic iron (Pereira et al., 2023a).

Some species were less specific to particular sources. $PO_4^{3-}$ presented no correlations with PMF factors and was moderately correlated with Na$^+$ and Ca$^{2+}$ ($r > 0.45$). Oxalate was not correlated with specific factors. It presented moderate correlations with $NH_4^+$, K$^+$, $NO_3^-$, $SO_4^{2-}$, As, Rb, Sb, Tl, Pb, and OC ($r > 0.5$). This suggests multiple sources for oxalate, as previously observed, such as vehicle exhaust, biomass burning, and biogenic activity, as well as secondary formation (Guo et al., 2010). Na$^+$ was not specific for any source, as previously observed in the MASP (Vieira-Filho et al., 2016b). This species presented a weak correlation with some other species (Ca$^{2+}$, Cl$^-$, $PO_4^{3-}$, As, Rb, Sr, and Cd) ($r > 0.3$). The lack of strong correlation with Cl$^-$ can be explained by the relatively low influence of sea spray in the studied period as indicated by the trajectories (Sect. 3.1). Aluminum displayed low correlations with the PMF factors and was strongly correlated with Ti ($r > 0.7$) and moderately with Ce ($r > 0.6$), pointing to its origin in crustal materials (Hetem and Andrade, 2016). Ti was also strongly correlated with Ce. Both species presented low enrichment (Sect. 3.6), suggesting a mineral origin apart from vehicular emissions. Chromium was not correlated with PMF factors but presented a relatively high correlation with Ni ($r \sim 0.5$). Both species were previously associated with industrial sources (Bourotte et al., 2011; Castanho and Artaxo, 2001). Arsenic, a highly toxic element, showed relatively moderate correlations with BB, IN, and VE2 and strong correlations with Rb, Ag, and Cd. Arsenic has multiple sources, including industries (Calvo et al., 2013).

## 3.5 Particle size distributions: associations with aerosol source apportionment and meteorological scenarios

Submicrometer particle number size distribution monitoring provided information about the relative contribution of different particle size modes and their temporal evolution. Total particle number concentrations averaged $1.03 \pm 0.59 \times 10^4$ cm$^{-3}$, with a mean geometric diameter of $51 \pm 16.1$ nm. Size distributions were dominated by the Aitken mode (mean geometric diameter range 50–100 nm) and the accumulation mode (diameter above 100 nm), respectively, accounting for 46 % and 32 % of particle number concentrations, on average. The nucleation mode (mean geometric diameter below 30 nm) was less frequent, contributing to an average of 21 % of the particle number concentration, on average (Table S5). Events of new particle formation and growth were infrequent, possibly due to the relatively high particle loading and condensational sink. Nearly 70 % of the

monitored particles had diameters below 100 nm, classified as ultrafine particles, which can penetrate the extensive area of the lungs and reach other organs through the lung vasculature (Schraufnagel, 2020). The predominance of Aitken mode particles suggests the influence of relatively fresh particles compared to the typically aged accumulation mode particles. A previous study in the MASP reported associations between inorganic species and the Aitken mode particles, whereas aged oxygenated organic particles were mainly associated with the accumulation mode (Monteiro dos Santos et al., 2021).

The size distribution measurements were averaged to match the filter sampling periods, providing a link with the aerosol chemical composition measurements and source apportionment. No significant differences were observed in the particle size distributions in days with a predominance of vehicular emissions or biomass burning. Although the aerosol chemical composition differed under the influence of different emission sources (Sect. 3.3), the particle size distributions were typically dominated by the Aitken mode, suggesting the presence of relatively fresh particles. However, there was a clear difference in the size distributions measured under high aerosol loadings (PM$_{2.5} \geq 15\,\mu g\,m^{-3}$, WHO guideline), with an increase of Aitken and accumulation mode particles, especially at the end of the day (Fig. S11). The atmospheric conditions associated with high PM concentration (low boundary layer height, weak ventilation, absence of precipitation, and clear-sky conditions) likely favored secondary aerosol production and particle growth by condensation in pre-existing particles (Sánchez-Ccoyllo and Andrade, 2002; Santos et al., 2018), which might explain the increase of larger particles (Monteiro dos Santos et al., 2021).

A period of 4 d TS15 was selected within the study period to assess the dynamics of particle size distribution in high temporal resolution and the influence of different meteorological scenarios. Figure 9a and b show the evolution of size distributions in 2 d with a strong contribution of secondary aerosol formation, according to the source apportionment analysis (Fig. 7). New particle formation events were observed on these days, with sub-30 nm particles emerging in the morning, followed by a consistent growth towards the Aitken mode until the mid-afternoon, with growth rates of 5.1 and 3.3 nm h$^{-1}$, respectively. The observed growth may be associated with the condensation of secondary organic and inorganic aerosol species. Previous studies in São Paulo indicated a large fraction of secondary inorganic species in the Aitken mode, likely produced from the reaction of nitric acid and ammonia (Carbone et al., 2013; Monteiro dos Santos et al., 2021). Another interesting feature was the entrance of the sea breeze around 16:00 LT on 15 August, marked by a dashed line in Fig. 9a. The sea breeze affected the measured size distributions, leading to a fast particle concentration decrease, from $1.8 \times 10^4$ to $1.0 \times 10^4\,cm^{-3}$, associated with a mean geometric diameter increase from $44.2 \pm 2.0$ to $47.0 \pm 2.0$ nm. The observed changes in the size distribution

may be explained by the transport of cleaner and humid air from the coast, likely associated with S and SE winds. In addition to oceanic air masses, aged particles from industrial emissions near the coast may also be transported to the MASP during sea breeze events.

On the other hand, Fig. 9c and d show examples of the influence of cold fronts on particle size distributions. The prefrontal period is typically characterized by stagnant atmospheric conditions that favor the occurrence of high PM$_{2.5}$ concentrations. The emergence of the cold front on the morning of 19 August TS16 resulted in increased ventilation and precipitation (Fig. S12), leading to a strong decrease in the particle number concentrations, from $2.8 \times 10^4$ to $0.5 \times 10^4\,cm^{-3}$. A decrease in the geometric mean diameter was simultaneously observed, from $94.7 \pm 2.0$ to $43.8 \pm 2.0$ nm, indicating that the particle removal by rain and winds was more efficient for accumulation mode particles. Figure 9 provides examples of sub-daily variability in aerosol properties, which cannot be discerned in the aerosol filter samples.

## 3.6 Chemical and source characteristics of pollution event days

The sampling days were separated between high-pollution days (with PM$_{2.5}$ exceeding the WHO guideline of $15\,\mu g\,m^{-3}$) and low-pollution days (below). These polluted periods were associated with relatively higher average temperatures and lower humidity (19 °C and 74 %, compared to 15 °C and 83 %). During these pollution events, PM$_{2.5}$ increased by 171 %, and carbonaceous species (OM+EC) represented a higher mass fraction (nearly 60 %) (Fig. 10a and b). In polluted periods, the accumulation processes can lead to the formation of organic coatings on black carbon particles (Monteiro dos Santos et al., 2021). On the other hand, oxides, other WSI, and sulfate accounted for a higher fraction in non-event days. The unidentified fraction, attributed to unmeasured components such as carbonate or water (Pereira et al., 2017a), was also slightly higher in non-event days. In absolute numbers, all PMF factors increased on event days. However, the BB factor contribution increased significantly on polluted days, rising from 4 % to 27 % of the PM$_{2.5}$ mass (Fig. 10c).

OC, EC, K$^+$, NO$_3^-$, V, Mn, Fe, Co, Zn, Rb, Cd, Sn, Sb, Ce, Tl, Pb, and Lev presented a significantly higher increase than that observed for PM$_{2.5}$ in these polluted periods (Table S6 and Fig. S13), suggesting an accumulation of these species during these events ($p < 0.05$). In percentage terms, all OC and EC fractions increased more than PM$_{2.5}$ ($p < 0.05$), except OC1, EC3, and EC4. Proportionally, retene increased less than PM$_{2.5}$, suggesting that it may be linked to a local biomass-burning source, such as wood burning in restaurants (Andrade et al., 2017; Kumar et al., 2016). The toxicity indexes BaP-TEQ and BaP-MEQ presented significant increases ($p < 0.05$), caused mainly by the rise in HMW-PAHs

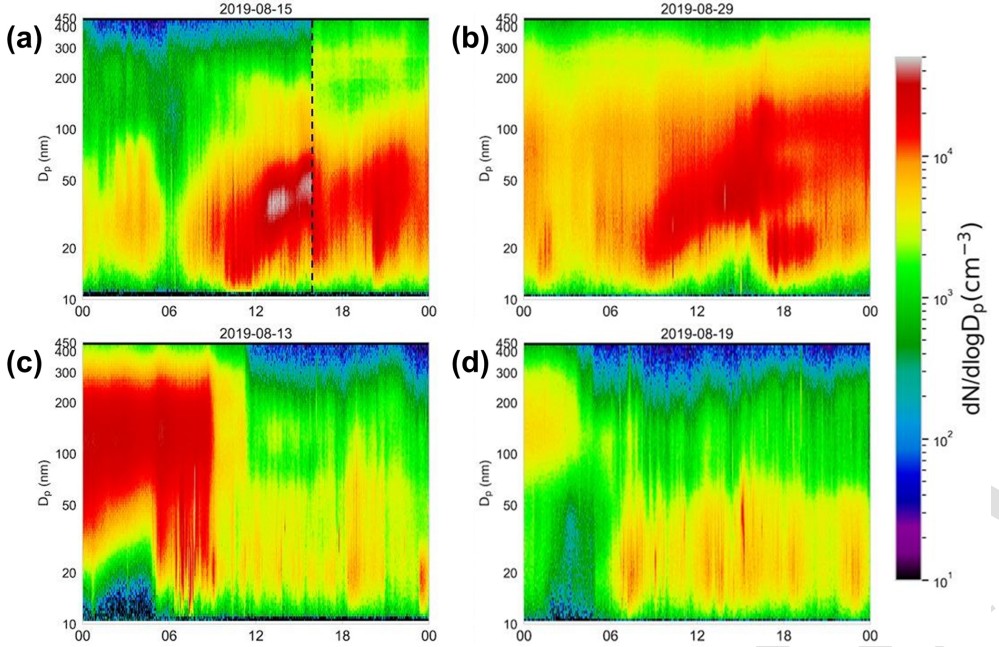

**Figure 9.** Submicrometer particle number size distributions for **(a)** 15, **(b)** 29, **(c)** 13, and **(d)** 19 August. The $x$ axis represents the timeline (local time), the $y$ axis refers to particle diameters, and the colors represent the number of particles normalized by diameter bins.

due to the accumulation of fossil-fuel and biomass-burning-emitted aerosols (Vasconcellos et al., 2010). Furthermore, there seems to be an accumulation of other toxic species from anthropogenic sources, such as potentially toxic elements (PTEs) such as Cd, Sb, and Pb (Pereira et al., 2023b; Thorpe and Harrison, 2008). However, the toxicity unity (TU) did not increase as much ($\sim 50\%$). Phosphate was the only species to decrease, although with $p > 0.05$. It is often associated with dust and farming activities (Yuan et al., 2008; Allen et al., 2010). In the present study, the most probable source was the soil.

Levoglucosan concentration tripled during the polluted days (423 and 140 ng m$^{-3}$, respectively). Ionic potassium quadrupled these days (330 and 82 ng m$^{-3}$), suggesting a higher impact of crop and other types of biomass burning (Chow et al., 2022). However, levoglucosan represented similar portions of OM on both occasions (nearly 3 %). Xylitol and mannosan presented a lower increase on polluted days ($p < 0.05$), suggesting they are associated with other types of biomass burning. In 2020, it was observed that during smoke plume events in the MASP, PM$_{2.5}$ surpassed 25 µg m$^{-3}$ at 99 % of the air quality monitoring stations. These days, the plumes were mainly associated with sugarcane burning and often with the contribution of wildfires in the Amazon and Pantanal biomes (Souto-Oliveira et al., 2023). In that study, the authors were able to differentiate CO$_2$ from local wood burning (4 %–7 %) from that from remote forest wildfires (4 %–26 %). In recent years, the influence of forest fires has grown, likely related to natural seasonal variations and climate change, with lower humidity and precipitation and in-

creased fires (Goss et al., 2020; Abram et al., 2021). The influence of biomass burning in the metropolitan area is often seen above the boundary layer. However, these plumes from remote regions can still influence the concentration and composition of PM$_{2.5}$ at the surface (Souto-Oliveira et al., 2023).

Enrichment factors for the elements were above 10 for Cr, Cu, Zn, As, Se, Mo, Ag, Cd, Sn, Sb, Ba, Tl, Pb, Bi, and U (Fig. 11), suggesting a strong anthropogenic character. Overall, there was an increase in the enrichment factors of elements in the polluted period, 5 times higher for Fe, suggesting that these are mostly soil-bound in the clean period. Despite the very low concentrations, Se, Sb, and Bi presented the highest EFs (near and above 10 000), indicating a very high anthropogenic character. Selenium was previously attributed to anthropogenic sources such as fossil fuel combustion, waste burning, tires and paper, coal combustion, oil, and glass industries (Mehdi et al., 2013). Additionally, it was previously attributed to industrial sources in the eastern part of the MASP (Vieira et al., 2023). Antimony is associated with non-exhaust emissions (brake wear) (Thorpe and Harrison, 2008) and non-ferrous-metal industries (Calvo et al., 2013). Bismuth is related to anthropogenic sources, such as fossil fuel combustion, metallurgy, and refuse incineration (Ferrari et al., 2000). Other species presented low EFs, attributed to geogenic sources: Ti, Fe, and Co (EFs near and below 1). Often associated with catalyst emissions (Kulkarni et al., 2006), La and Ce displayed relatively low enrichment factors (below 5). Some species exhibited a geogenic character in the clean periods and a more anthropogenic character in the polluted period, with K, V, Ni, and Rb approaching an EF of 10.

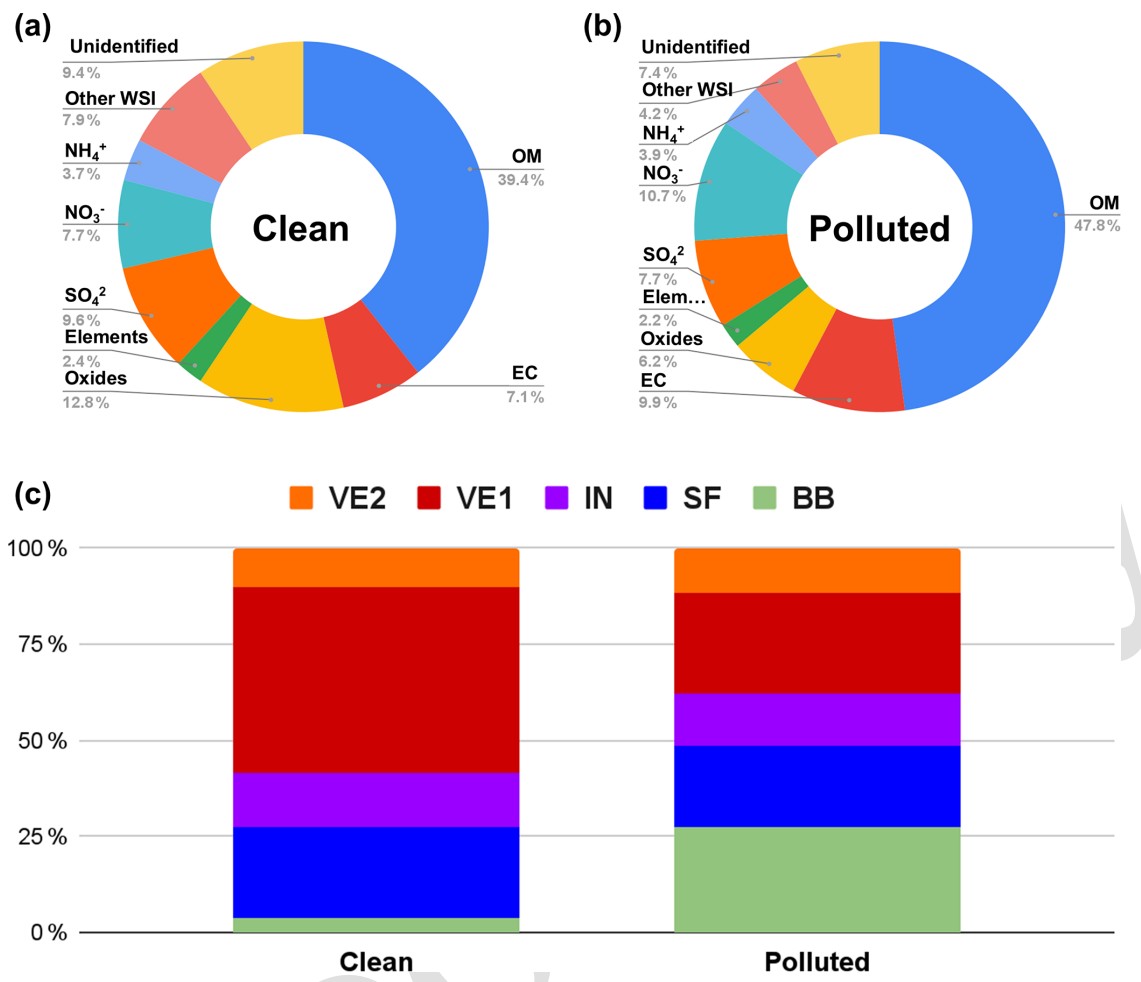

**Figure 10.** Chemical mass balance for clean days ($PM_{2.5} < 15\,\mu g\,m^{-3}$) **(a)** and polluted days ($PM_{2.5} \geq 15\,\mu g\,m^{-3}$) **(b)**, as well as PMF source contributions for both periods **(c)**.

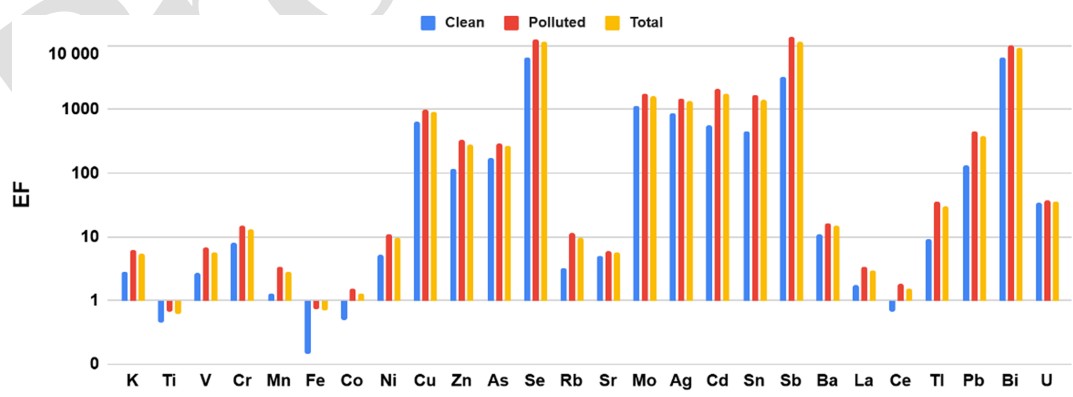

**Figure 11.** Enrichment factors (EFs) for the clean, polluted, and total period.

## 3.7 Ecotoxicity assays

Ecotoxicity tests of aqueous particulate matter extracts were performed with the bacteria *Aliivibrio fisheri*. All samples in this study were classified as toxic, with toxic unity (TU) values ranging from 1.7 to 7.1, averaging 3.7. The highest values were obtained in periods impacted by biomass burning (Fig. S14). The TU unit showed moderate correlations with levoglucosan, water-soluble $K^+$, and the modeled aqueous $K^+$ ($r >$ TS17 0.5), which may be associated with

biomass-burning aerosols and water-soluble organic species (not quantified in the study) (Urban et al., 2012). Among the factors obtained by receptor modeling, biomass burning had the highest correlation with TU (Table S4), indicating that water-soluble components trigger the toxicity of particles emitted by this source. The local vehicular factor exhibited the second highest correlation with TU, although weak. In addition, the TU values presented moderate correlations with elements known to be toxic and associated with vehicular sources such as Pb ($r \sim 0.5$). The correlation was moderate ($r > 0.5$) with OC (higher for fractions OC2, OC3, and OC4, less volatile and more oxidized) and EC (higher for fractions EC1 and EC2). Significant correlations between the bioluminescence inhibition responses and the contributions of biomass burning and traffic to particulate matter concentrations were also reported for Coimbra, Portugal (Alves et al., 2021). As observed in the present study, toxicity was statistically correlated with OC, EC, anhydrosugars, and elements from exhaust and non-exhaust emissions. Since biomass-burning and traffic emissions may elicit acute toxic effects, adopting source-specific preventive and remedial measures is necessary.

## 4 Summary and conclusions

Fine particulate matter (PM$_{2.5}$) was collected in a 100 d dry period in 2019, covering the period from June to early September, when several pollution events were observed, surpassing the WHO daily limit of $15 \mu g \, m^{-3}$ in 75 % of the days. Chemical characterization was obtained, including water-soluble ions, elements, carbonaceous species, anhydrosugars, and polycyclic aromatic hydrocarbons. Ecotoxicity was assessed using a bioluminescence-based assay. Additionally, the size distribution of particles (SMPS) was monitored simultaneously. The lower sulfate-to-nitrate ratios suggested a decrease in sulfur oxide levels. A higher contribution of organic matter to particulate matter indicated an increase in the secondary formation of organic species and a reduction of elemental carbon emissions by vehicles. However, a further study should be performed to statistically assess this trend over the last decades. The impact of oxidant atmosphere and higher temperatures on the formation of secondary organic carbon and the effect of industrial emissions on sulfate production require further investigation.

As for biomass burning, typically observed in the dry period, there was a decrease in the contribution of sugarcane straw burning since there was a change in the K$^+$ / Lev and Lev / Man ratios and relatively lower potassium levels. However, the long-range transport of plumes from forest fires and agricultural burning in regions north and northwest of MASP remains a significant source of PM$_{2.5}$ in this period, as the average concentration of levoglucosan remained high. Other authors have observed a further increase in the influence of bagasse-burning plants and the intrusion of aerosols from forest fires originating in central and northern Brazil. This last phenomenon is related to climate change and has increased in the last decades (longer dry periods). The increase in the Lev / Man ratio with stronger winds suggests the contribution of different types of biomasses. Correlations with other species and the PMF receptor model suggested Rb as a reliable biomass-burning tracer. Furthermore, PAHs, which are often found in elevated concentrations in the dry period, remain a concern due to equivalent toxicity values exceeding 1 ng m$^{-3}$ in half of the samples. However, the levels are lower than those observed in previous studies. The vehicular-related species BbF remained an abundant PAH, suggesting it is a persistent constituent. Furthermore, PAH diagnostic ratios fell within the range observed for vehicular emissions. The increased concentrations of these pollutants are likely related to the lower dispersion in this season. Additionally, increasing V / Ni and La / Ce ratios with east and southeast winds suggested a contribution of aerosols from petrochemical industrial areas, which can occur with meteorological conditions characterized by cold fronts and sea breezes.

The PMF receptor model was applied to assess the PM$_{2.5}$ sources, and a five-factor solution was obtained (biomass burning, secondary formation, industrial, vehicular+road dust, and local vehicular). A high contribution of biomass burning, associated with north and northwest winds, was observed, reaching one-fourth of the particulate matter. Considering the previous source apportionment studies, sources related to vehicular emissions are still dominant (40 % of PM$_{2.5}$). A mixed factor of road dust and vehicular emissions increased throughout the campaign, suggesting a more significant influence of resuspension at the end of the winter. The industrial contribution was relatively lower, increasing with northeast winds that pass through industrial areas of MASP. The PMF solution showed overlapping contributions in some factors, which may be related to the low temporal resolution of sampling and the fact that emissions from various sources mix before reaching the semi-background receptor site. Enhancing the time resolution in future investigations may help identify more sources.

Two particle formation events were identified by SMPS in days with pronounced secondary formation and happened before the arrival of the sea breeze. The sulfate secondary formation was related to humid conditions, as suggested by correlations between the contribution of secondary formation and the aerosol's modeled water content (ISORROPIA). During days of pollution events (PM$_{2.5}$ > $15 \mu g \, m^{-3}$), carbonaceous species represented a higher fraction of particulate matter, while sulfate's contribution was reduced. An accumulation of PAHs and toxic species such as Cd, Sb, and Pb on these days represents a health concern. These pollution events were associated with a relative increase in the contribution of the biomass-burning factor, whose emissions, added to pollutants emitted and formed locally, contribute to the degradation of air quality in the dry season. Throughout the sampling campaign, all samples were classified as eco-

toxic. The ecotoxicity correlated with the biomass-burning factor, highlighting the importance of regulating this source for air quality control. These results indicate that controlling PM$_{2.5}$ exceedance events should include regulating emerging biomass-burning sources and stricter rules concerning vehicular emissions.

**Data availability.** The datasets is TS18 available upon request. TS19

**Supplement.** The supplement related to this article is available online at [the link will be implemented upon publication].

**Author contributions.** GMP performed part of the analyses (OC, EC, elements, and water-soluble ions), performed the data treatment, and led the manuscript writing. MFA managed the project, provided infrastructure, designed the study, and revised the manuscript. PCV provided infrastructure, designed the study, and revised the manuscript. LYK performed the data treatment. RFB monitored the particle size distribution and contributed to the writing. DMS helped with the OC and EC determination and revised the manuscript. TSS performed the PAH extraction and characterization. CG characterized the anhydrosugars. ICR performed the ecotoxicity assays and helped with the revisions. CA, NK, LR, PA, and EDF provided infrastructure and revised the manuscript. TN, RMM, and MAY revised the paper.

**Competing interests.** The contact author has declared that none of the authors has any competing interests.

**Disclaimer.** Publisher's note: Copernicus Publications remains neutral with regard to jurisdictional claims made in the text, published maps, institutional affiliations, or any other geographical representation in this paper. While Copernicus Publications makes every effort to include appropriate place names, the final responsibility lies with the authors.

**Acknowledgements.** These results were part of the Chemical and toxicological SOurce PROfiling of particulate matter in urban air project (SOPRO, grant no. POCI-01-0145-FEDER-029574), supported by the Portuguese Foundation for Science and Technology from the Ministry of Science, Technology, and Higher Education (FCT/MCTES) and by the São Paulo Research Foundation (FAPESP; grant no. 2018/07848-9). The SOPRO project was also funded by FEDER, through COMPETE2020 – Programa Operacional Competitividade e Internacionalização (POCI), and by national funds (OE), through FCT/MCTES. The authors also acknowledge FAPESP for grant no. 2016/18438-0 (Metroclima project, for research resources) and grant no. 2019/01316-8 (URBESP project, scholarship), as well as National Research Council (grant no. CNPq 301503/2018-4). The financial support to CESAM by FCT/MCTES (UID Centro de Estudos do Ambiente e Mar (CESAM) + LA/P/0094/2020), through national funds, is also acknowledged. The authors also acknowledge the framework INCT Klimapolis, funded by CNPq TS20 (grant no. 406728/2022-4). Meteorological data (irradiance, atmospheric pressure, temperature, and relative humidity) were kindly provided by the IAG-USP meteorological station and the Climatology and Biogeography Laboratory (Department of Geography, FFLCH, USP).

**Financial support.** This research has been supported by the Fundação de Amparo à Pesquisa do Estado de São Paulo (grant nos. 2018/07848-9, 2016/18438-0, and 2019/01316-8), the Fundação para a Ciência e a Tecnologia (grant nos. POCI-01-0145-FEDER-029574 TS21; UID (CESAM) + LA/P/0094/2020 TS22), and the Conselho Nacional de Desenvolvimento Científico e Tecnológico (grant no. 301503/2018-4).

**Review statement.** This paper was edited by Dara Salcedo and reviewed by two anonymous referees.

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

## Remarks from the language copy-editor

CE1    Please confirm the changes to this caption.

CE2    Please provide an explanation for the editor for the requested change: "Fewer than 10 % of the samples presented ratios above 0.4, which may be associated with relatively fresher emissions."

## Remarks from the typesetter

TS1    Please note that a space between the number an the percent symbol is one of our standards an cannot be removed.

TS2    Please note that it is our standard to format dates like this; therefore, the requested change could not be inserted.

TS3    Please note that st, nd, rd and th are not superscripted as per our standards.

TS4    Please note that it is our standard to insert skinny spaces instead of commas in numbers exceeding four digits.

TS5    Please note that stylistic changes as requested for Table 1 are not possible at this stage.

TS6    Please note that the size of Fig. 3 cannot be reduced as the text is already now barely legible.

TS7    Please correct "Meng et al., 2020". No exact match was found in the reference list.

TS8    Please note that the size of Fig. 5 cannot be reduced as some of the content is already now barely legible.

TS9    Due to the requested changes, we have to forward your requests to the handling editor for approval. To explain the corrections needed to the editor, please send me the reason why these corrections are necessary. Please note that the status of your paper will be changed to "Post-review adjustments" until the editor has made their decision. We will keep you informed via email.

TS10    Please note that stylistic changes like this are not possible at this stage.

TS11    Please provide a corresponding footnote for "1". If the same content is meant as in the footnote on page 10, please note that the entire footnote needs to be repeated. Just repeating the number is technically not possible.

TS12    Please provide a corresponding footnote for "2". If the same content is meant as further below in the text, please note that it needs to be repeated.

TS13    Please provide a corresponding footnote for "2". If the same content is meant as further below in the text, please note that it needs to be repeated.

TS14    Please note that changes like the requested additions and the other changes to $r$ and $p$ values are generally not possible at this stage without editor approval. In case these changes are essential, we need to forward them to the editor in the "Post-review adjustments" process as explained above. To do this, we would need a detailed explanation of each change.

TS15    Please note that it is our standard to abbreviate SI-accepted units in combination with numbers, including d for "day".

TS16    Thank you for catching this.

TS17    Please see previous remarks regarding editor approval.

TS18    Please confirm.

TS19    You wrote that "The openair package was generically cited.". If this package was used for this paper, please provide a corresponding statement in the Code and data availability section including a direct link/DOI to the code and the corresponding reference list entry (including creators, title, repository/publisher, and date of last access).

TS20    Please also add this information to the "Financial support" section.

TS21    This grant number was kept as it is also still listed in the Acknowledgements

TS22    Is LA/P/0094/2020 a grant number? If so "grant no." should be added.

TS23    Please check this reference. A matching citation was not found.

TS24    Please provide publisher.

TS25    Please confirm article number.