# Peer review of "Source apportionment and ecotoxicity of PM2.5 pollution events in a"

_EGUsphere, 2024_

## Referee Comment (RC2)

**Comments for manuscript:** "Source apportionment and ecotoxicity of particulate pollution events in a Major Southern Hemisphere Megacity: influence of biomass burning and a biofuel impacted fleet" by Guilherme Martins Pereira.

The manuscript investigates the chemical composition of fine particulate matter ($PM_{2.5}$) during a 100-day dry period in 2019 in the Metropolitan Area of São Paulo (MASP), Brazil. Source apportionment using Positive Matrix Factorization highlights vehicular emissions and biomass burning as the dominant $PM_{2.5}$ sources. While the main text is well-written and presents compelling results, the data comparisons are primarily limited to previous studies in São Paulo. Expanding comparisons to other major cities or continents could provide broader context and enhance the study's contribution to global air quality research.

The title should include $PM_{2.5}$ and focus firstly on vehicular emissions and afterwards on biomass burning related influence. The figures could benefit from some revisions (see comments below).

**Major Comments:**

**Line 228**: Could you provide an explanation for why nitrate concentrations surpassed sulfate during this campaign? Are there any measurements of sulfur oxides (e.g., from CETESB) over the last decade that support a reduction in sulfate levels?

**Line 283**: You present back-trajectories for two specific days, describing them as representative of typical air mass influences. Could you include the percentage of air masses arriving from this direction over the entire sampling period? Additionally, please describe the source regions for the remaining air masses during the campaign. Consider including a figure summarizing HYSPLIT-derived air mass sources and their percentages, either in the main text or the Supplementary Information (SI).

**Line 290**: Could you elaborate on xylitol? Why is it of particular interest? How do its concentrations compare to previous campaigns/sites? Are its levels higher during high or low relative humidity? Is there any relationship between xylitol levels and specific air mass orientations?

**Line 318**: Please provide the OC/EC ratio for the entire campaign in the SI. Is the observed increase in the OC/EC ratio during this campaign due to an increase in OC, a decrease in EC, or a combination of both?

**Line 329**: Are ozone ($O_3$) measurements from 2019 available to support your observations? If so, please include relevant data.

**Line 369**: Could you clarify what is meant by the "dark" precipitation event? It is clearly not a typical rain event, so this term should be described earlier in the manuscript. Additionally, $PM_{2.5}$ concentrations during these days are reported close to 10 µg/m$^3$, whereas values below 5 µg/m$^3$ might be expected. Please explain this discrepancy. PMF may help you here.

**Temperature and RH Time Series**: Could you include the time series for temperature and relative humidity during the measurement period in the SI? The averages are mentioned somewhere in the text but the information is lost.

**Figure 6**: Could you clarify what is meant by "reference runs" and "BS"?

**PMF Analysis, Section 3.4**:

What is the correlation between the identified factors? For example, does VE1 increase after VE2? To support the resuspension? Providing more details on these relationships would enhance understanding.

**Section 3.7 Placement**: Consider moving Section 3.7 after section 3.4, as this may improve the manuscript's flow.

**Global Comparison of PMF Analysis**: While PMF analysis on $PM_{2.5}$ is discussed for MASP, could you expand on comparisons with findings from other global studies? For example, comparisons to Srivastava et al. (2021), Wang et al. (2018), Han et al. (2023), Cheong et al. (2024), Nava et al. (2020), and others could provide broader insights.

**Abstract and Figure 7 Discrepancy**: In the abstract, you state that "sources related to vehicular emissions remain dominant (over 60% of $PM_{2.5}$). However, Figure 7 shows VE1 and VE2 contributing approximately 40%. Could you clarify why secondary formation (SF) factor is included with car-related emissions? Additionally, Line 452 relates this SF factor to industrial emissions rather than vehicular ones. Moreover, in Section 3.7, the SF factor is described as having a weak correlation with $NO_3^-$/EC and an opposite trend to the BB, VE1, and VE2 factors. Please reconcile these points.

**Lines 637–644**: Nickel and vanadium are also recognized as tracers for ship exhaust (Zhao et al., 2021). Have you considered this possibility in your analysis?

**VE1 and VE2 vs. Heavy- and Light-Duty Vehicles**: A previous study (Vieira et al., 2023) identified separate factors for heavy- and light-duty vehicles. Could you explain how VE1 and VE2 in your study differ from or align with these factors?

**Marine-Related Factor and 6-Factor Solution**: On August 15 and 29, a sea breeze was observed. Could a marine-related factor be present? Additionally, would a 6-factor solution help better deconvolute the sources? Some factors in Section 3.4 appear to be mixtures of two or more sources, which a 6-factor approach might resolve. Comparison to global studies will shed light on this study's findings.

**Pollution Events and Particle Size Distribution**: This section could benefit from revision to provide greater depth. The first part reads like an introductory paragraph, while the second discusses four specific days without adequately explaining their significance. Additionally, the entire measurement period is only briefly addressed. Including a table summarizing average values (e.g., new particle formation rates, growth rates, nucleation event days) in the main manuscript would enhance clarity and aid the reader.

**Enrichment Factor (EF) Results**: Although the methodology for calculating enrichment factors is described in the data treatment section, no corresponding results, tables, or figures are provided in the main text. Apart from the brief mention in Lines 610–611, could you include these results to strengthen the manuscript?

**Minor Comments:**

**Line 89.** Replace the word "extended"

**Line 118.** Rewrite the phrase "the first...derivatives". The second step looks like is missing something.

**Lines 157-159.** Replace "BaPTEQ" to "BaP$_{TEQ}$", "BaPMEQ" to "BaP$_{MEQ}$", "BaPEq" to "BaP$_{Eq}$". Fix/increase the "(2)" to be the same as the others. Explain in what units the BaA, BkF etc. are referring to.

**Line 177.** Add reference after "depleted". Replace "It" with "EF". Add "with" before EF.

**Lines 179-180.** Either use parentheses for all CXp etc., or for none of the four. I recommend you use parentheses for none. Also, from which study do you take the concentration in the Earth's crustal material? Please add references.

**Line 251.** Please present Figure 2 in order, for example not first the Fig. 2e and then Fig. 2c-d. Change the order of chart pies in Figure 2 to be consistent with the manuscript.

**Lines 252-256.** What are EC1, EC2, OC1, OC2 etc. are referring to? Please explain.

**Figure 2.** Increase the fonts for Lev, Lev/Man, PM2.5, EC and OM. Add to the caption that the time series in a and b refer to daily data. What do you mean by "dark" rain event?

**Line 276.** Add reference of previous campaign (2014).

**Figure 4.** Increase fonts. Increase the graph to be clearer to the reader. It would be nice to add the annual limit recommended by the European Environment Agency with different color.

**Figure 5**: Could you provide additional details in the main text on how these polar plots were generated?

**Line 362.** The number "10" and "9" it is not very clear to what they are referring to.

**Line 364.** Same for number "19"

**Line 367.** Maybe wanted to write "Figure 2" instead of "Figure 1"?

**Line 434**. Mention the factors by the order you present those in Figure 6. It's easier for the reader.

**Figure 6.** Increase all fonts (axis, species, Factors). Especially, the "y-axis" cannot be read. Also, you could use a common x-axis in the bottom of the figure and increase the bars (maybe use different colors for each factor).

**Line 457.** Move "Factor 3" to another paragraph.

**Figure 7.** Better use µg m$^{-3}$ instead of ng m$^{-3}$. Increase fonts. Add dark rain event purple bar.

**Figure 8.** In the caption there is a "b)" in the end of the sentence without anything following. Maybe move a) the b) before "trajectory" and "participation", respectively.

**Line 541.** Remove the double dot.

**Figure 9.** Increase all fonts. At the caption, move (a)-(d) before the dates.

**Line 561.** Units for number concentration?

**Lines 562-563.** "The distribution, which had a geometric mean diameter of 94.7 ± 2.0 nm at 8 AM, shifted to particles with a mean size of 43.8 ± 2.2 nm." And? What is the explanation for this? The key point?

**Line 570.** A dot is missing after "…83%)"

**Section 3.6:** A Table with the polluted/ clean concentration values for all species would be an interesting addition.

**References:**

Cheong, Y., Kim, T., Ryu, J. *et al*. Source apportionment of PM$_{2.5}$ using DN-PMF in three megacities in South Korea. *Air Qual Atmos Health* (2024). https://doi.org/10.1007/s11869-024-01584-5.

Yun-Sung Han, Da-Mee Eun, Greem Lee, Sung Yong Gong, Jong-Sang Youn, Enhancement of PM2.5 source appointment in a large industrial city of Korea by applying the elemental carbon tracer method for positive matrix factorization (PMF) model, Atmospheric Pollution Research, Volume 14, Issue 11, 2023, 101910, ISSN 1309-1042, https://doi.org/10.1016/j.apr.2023.101910.

Nava, S.; Calzolai, G.; Chiari, M.; Giannoni, M.; Giardi, F.; Becagli, S.; Severi, M.; Traversi, R.; Lucarelli, F. Source Apportionment of PM$_{2.5}$ in Florence (Italy) by PMF Analysis of Aerosol Composition Records. *Atmosphere* **2020**, *11*, 484. https://doi.org/10.3390/atmos11050484.

Srivastava, D., Xu, J., Vu, T. V., Liu, D., Li, L., Fu, P., Hou, S., Moreno Palmerola, N., Shi, Z., and Harrison, R. M.: Insight into PM$_{2.5}$ sources by applying positive matrix factorization (PMF) at urban and rural sites of Beijing, Atmos. Chem. Phys., 21, 14703–14724, https://doi.org/10.5194/acp-21-14703-2021, 2021.

Vieira, E. V. R., do Rosario, N. E., Yamasoe, M. A., Morais, F. G., Martinez, P. J. P., Landulfo, E. and Maura de Miranda, R.: Chemical characterization and optical properties of the aerosol in São Paulo, Brazil, Atmosphere, 14(9), 1460, https://doi.org/10.3390/atmos14091460, 2023.

Wang Q.Q., L.P. Qiao, M. Zhou, S.H. Zhu, S. Griffith, L. Li, J.Z. Yu Source Apportionment of PM$_{2.5}$ Using Hourly Measurements of Elemental Tracers and Major Constituents in an Urban Environment: investigation of Time-Resolution Influence J. Geophys. Res.-Atmos., 123 (2018), pp. 5284-5300.

Zhao, J., Zhang, Y., Xu, H., Tao, S., Wang, R., Yu, Q., Chen, Y., Zou, Z., Ma, W., 2021. Trace elements from ocean-going vessels in east Asia: vanadium and nickel emissions and their impacts on air quality. J. Geophys. Res. Atmos. 126, 1–16. https://doi.org/ 10.1029/2020JD033984.

---

## Author Comment (AC1)

**Response to reviewers' comments**

The authors thank both reviewers for their insightful comments, which helped in the improvement of this manuscript. We have taken them into account and revised the manuscript. The indicated lines refer to the marked manuscript. We have replaced the figures with higher resolution versions. Figures and tables were moved to more appropriate positions in the text. Supplementary Information was reformulated, as well. Additionally, Section 3.5 has been rewritten, and Section 3.7 was relocated to follow Section 3.3. Below, you will find our responses to your comments, highlighted in blue.

**Reviewer #2**

**C1:** The manuscript investigates the chemical composition of fine particulate matter (PM2.5) during a 100-day dry period in 2019 in the Metropolitan Area of São Paulo (MASP), Brazil. Source apportionment using Positive Matrix Factorization highlights vehicular emissions and biomass burning as the dominant PM2.5 sources. While the main text is well-written and presents compelling results, the data comparisons are primarily limited to previous studies in São Paulo. Expanding comparisons to other major cities or continents could provide broader context and enhance the study's contribution to global air quality research.

**A1:** New references have been incorporated throughout Results and Discussions for comparisons, especially on section 3.4, as detailed in the following responses to these comments.

**C2:** The title should include PM2.5 and focus firstly on vehicular emissions and afterwards on biomass burning related influence. The figures could benefit from some revisions (see comments below).

**A2:** The authors thank the reviewer for the suggestion. Title was reformulated: "Source apportionment and ecotoxicity of PM2.5 pollution events in a Major Southern Hemisphere Megacity: influence of a biofuel impacted fleet and biomass burning"

Major Comments:

**C3:**Line 228: Could you provide an explanation for why nitrate concentrations surpassed sulfate during this campaign? Are there any measurements of sulfur oxides (e.g., from CETESB) over the last decade that support a reduction in sulfate levels?

**A3:** The concentrations of sulfur dioxide (SO2) measured by the CETESB network in the Metropolitan Area of São Paulo (MASP) have decreased significantly, from 16  $\mu$ g m-3 in 2000 to 2  $\mu$ g m-3 in 2023 (added in the SI as Figure S3). This steady decline can be attributed to stricter regulations on sulfur content in fuels such as diesel and gasoline, as well as a shift in industrial energy sources from oil-fired boilers to natural gas and electricity, the discussion was updated in lines 270-276.

The graphs presented in the CETESB report in 2019 illustrate a more pronounced decrease in  $SO_2$  levels compared to nitrogen dioxide ( $NO_2$ ) levels during a similar period (2010 to 2019). This trend is further supported by lower estimated emissions of  $SO_2$  when compared to NOx. The reduction in sulfur dioxide emissions has also been discussed in the paper by Andrade et al. (2017 - Atmos. Environ., 159, 66–82), which is cited in the manuscript.

The observed concentrations of  $SO_2$  (averaged across all sites in the MASP) are shown in graph (a), while graph (b) displays  $NO_2$  concentrations from several sites within the MASP (these figures were extracted from the CETESB report(1) and will not be included in the manuscript or supplementary material):

Additionally, estimates of emissions for NOx and SO2 are represented in the graphs, with emissions measured in thousands of tons on the y-axis (these figures were extracted from the CETESB report(1) and will not be included in the manuscript or supplementary material):

---

## Author Comment (AC3)

**Response to reviewers' comments**

The authors thank both reviewers for their insightful comments, which helped in the improvement of this manuscript. We have taken them into account and revised the manuscript. The indicated lines refer to the marked manuscript. We have replaced the figures with higher resolution versions. Figures and tables were moved to more appropriate positions in the text. Supplementary Information was reformulated, as well. Additionally, Section 3.5 has been rewritten, and Section 3.7 was relocated to follow Section 3.3. Below, you will find our responses to your comments, highlighted in blue.

**Reviewer #2**

**C1:** The manuscript investigates the chemical composition of fine particulate matter ($PM_{2.5}$) during a 100-day dry period in 2019 in the Metropolitan Area of São Paulo (MASP), Brazil. Source apportionment using Positive Matrix Factorization highlights vehicular emissions and biomass burning as the dominant $PM_{2.5}$ sources. While the main text is well-written and presents compelling results, the data comparisons are primarily limited to previous studies in São Paulo. Expanding comparisons to other major cities or continents could provide broader context and enhance the study's contribution to global air quality research.

**A1:** New references have been incorporated throughout Results and Discussions for comparisons, especially on section 3.4, as detailed in the following responses to these comments.

**C2:** The title should include $PM_{2.5}$ and focus firstly on vehicular emissions and afterwards on biomass burning related influence. The figures could benefit from some revisions (see comments below).

**A2:** The authors thank the reviewer for the suggestion. Title was reformulated: "Source apportionment and ecotoxicity of $PM_{2.5}$ pollution events in a Major Southern Hemisphere Megacity: influence of a biofuel impacted fleet and biomass burning"

Major Comments:

**C3:** Line 228: Could you provide an explanation for why nitrate concentrations surpassed sulfate during this campaign? Are there any measurements of sulfur oxides (e.g., from CETESB) over the last decade that support a reduction in sulfate levels?

**A3:** The concentrations of sulfur dioxide ($SO_2$) measured by the CETESB network in the Metropolitan Area of São Paulo (MASP) have decreased significantly, from 16 µg m$^{-3}$ in 2000 to 2 µg m$^{-3}$ in 2023 (added in the SI as Figure S3). This steady decline can be attributed to stricter regulations on sulfur content in fuels such as diesel and gasoline, as well as a shift in industrial energy sources from oil-fired boilers to natural gas and electricity, the discussion was updated in lines 270-276.

The graphs presented in the CETESB report in 2019 illustrate a more pronounced decrease in $SO_2$ levels compared to nitrogen dioxide ($NO_2$) levels during a similar period (2010 to 2019). This trend is further supported by lower estimated emissions of $SO_2$ when compared to NOx. The reduction in sulfur dioxide emissions has also been discussed in the paper by Andrade et al. (2017 - Atmos. Environ., 159, 66–82), which is cited in the manuscript.

The observed concentrations of $SO_2$ (averaged across all sites in the MASP) are shown in graph (a), while graph (b) displays $NO_2$ concentrations from several sites within the MASP (these figures were extracted from the CETESB report[1] and will not be included in the manuscript or supplementary material):

[Figure]

**(a)**

[Figure]

**(b)**

Additionally, estimates of emissions for NOx and SO$_2$ are represented in the graphs, with emissions measured in thousands of tons on the y-axis (these figures were extracted from the CETESB report[1] and will not be included in the manuscript or supplementary material):

[Figure]

Reference:

(1) CETESB: Companhia de Tecnologia do Saneamento Ambiental: Relatório de qualidade do ar no Estado de São Paulo 2019, Report of air quality in the São Paulo State 2019, São Paulo, Brazil, available at: http://ar.cetesb.sp.gov.br/ publicacoes-relatorios/, 2020.

**C4:** Line 283: You present back-trajectories for two specific days, describing them as representative of typical air mass influences. Could you include the percentage of air masses arriving from this direction over the entire sampling period? Additionally, please describe the source regions for the remaining air masses during the campaign. Consider including a figure summarizing HYSPLIT-derived air mass sources and their percentages, either in the main text or the Supplementary Information (SI).

**A4:** The authors thank the reviewer for the suggestion. The trajectory clusters were calculated and inserted in Figure 3 (line 322). These clusters indicate that most trajectories passing over the continent originate from the northwest, where crop burning and wildfires are typically observed (considering a height of 500 m; see Figure (a) below). Less than 20% of the trajectories come from the ocean, which may explain the low correlation between sodium and chloride during this period. A discussion on this topic has been included in lines 311-314. Additionally, around 20% of the trajectories pass through the

northeastern region of the MASP, traversing industrial areas and a significant portion of the metropolitan area.

[Figure]

**(a)**                                                                                    **(b)**

**C5:** Line 290: Could you elaborate on xylitol? Why is it of particular interest? How do its concentrations compare to previous campaigns/sites? Are its levels higher during high or low relative humidity? Is there any relationship between xylitol levels and specific air mass orientations?

**A5:** The authors thank the reviewer for the suggestion. This species can present different sources and has been quantified across different sampling sites recently. Xylitol was a predominant polyol in the aerosols collected in this campaign. As mentioned in lines 325-327: "is mainly found in biological material, soil biota, and biomass-burning smoke (Marynowski and Simoneit, 2022; Gonçalves et al., 2021; Caseiro et al., 2007)."

A comparison with previous studies was inserted in lines 327-330: "Gonçalves et al. (2021) found a relation between this polyol and biomass-fuelled heating and reported an increase from 2.95 ng m$^{-3}$ in summer to 19.9 ng m$^{-3}$ in winter, in an urban background site in Portugal. although, much lower concentrations were observed in an urban site near sugarcane plantation areas in the countryside of São Paulo (below 2 ng m$^{-3}$) (Carvalho et al., 2023)."

In the Section 3.4 (Correlations between PMF results and other variables), it was mentioned that "xylitol was weakly correlated with BB and VE1 (r > 0.3). Notably, it presented weaker correlations with potassium than Lev, suggesting a biomass-burning origin less associated with crop burning". Additionally, a weak negative correlation with RH was observed, indicating that xylitol concentrations tend to increase with lower humidity, it was updated in lines 637-638.

We analyzed trajectory frequencies for the days with the highest xylitol concentrations (August 29 and 30) at altitudes of 500 m (a) and 3000 m (b) above ground level. On August

30, concentrations reached up to 75 ng m³, with the trajectories passing through regions that registered biomass burning. It was added to the discussions in lines 330-333 and SI (Figure S5).

[Figure]

(a)                                                        (b)

**C6:** Line 318: Please provide the OC/EC ratio for the entire campaign in the SI. Is the observed increase in the OC/EC ratio during this campaign due to an increase in OC, a decrease in EC, or a combination of both?

**A6:** The figure was included in the supplementary information (SI) as Figure S7 item (a).

The increase in the organic carbon to elemental carbon (OC/EC) ratio is likely the result of two factors: a relative increase in organic carbon due to a rise in secondary organic carbon formation, alongside a reduction in elemental carbon emissions, as detailed in the excerpt from lines 369-372:

"The ratio increase between these years may indicate a more significant influence of biofuel consumption and reduction of EC emissions (Pereira et al., 2023a), since the ethanol sales in the state of SP in 2019 were 75% higher than in 2013 (MME, 2023). Furthermore, an enhanced contribution from secondary organic carbon (SOC) formation may also contribute to this. SOC was estimated to reach nearly half of OC, with a maximum of 83% (Table 1)."

In lines 378-379, it was added that "The enhancement of SOC under a more oxidative atmosphere requires further investigation."

**C7:** Line 329: Are ozone ($O_3$) measurements from 2019 available to support your observations? If so, please include relevant data.

**A7:** In recent years, concentrations of most primary pollutants ($SO_2$, CO, and $PM_{10}$) have decreased. However, there has been a slight increase in ozone levels (Andrade et al., 2017). It is important to note that ozone concentrations can fluctuate from year to year; check the figure from the CETESB report[1] below for ozone yearly evolution:

[Figure]

In São Paulo, the highest ozone concentrations typically occur in spring rather than the summer. This is because summer is often characterized by increased cloud cover and frequent rainfall, which limit ozone formation. In fact, an increase in ozone concentration was observed at a CETESB monitoring station located on the university campus (IPEN) towards the end of the campaign as spring approached. Additionally, a moderate correlation (r ~ 0.45) was found between $O_3$ and OC/EC levels. Notably, peaks in OC/EC on July 28, August 3, 11, 18, and 25 coincided with peaks in ozone concentrations.

[Figure]

However, the relationships between these factors are not definitive and require further investigation (lines 378-379).

**C8:** Line 369: Could you clarify what is meant by the "dark" precipitation event? It is clearly not a typical rain event, so this term should be described earlier in the manuscript. Additionally, PM2.5 concentrations during these days are reported close to 10 µg/m³, whereas values below 5 µg/m³ might be expected. Please explain this discrepancy. PMF may help you here.

**A8:** The dark precipitation event is now described in detail in the methodology (lines 91-92). On August 19 and the day following the event, $PM_{2.5}$ concentrations ranged from 7 to 9 µg m$^{-3}$. These levels were relatively lower than those observed prior to the event and represented some of the lowest concentrations recorded. This decrease occurred after the arrival of the cold front. Notably, daily values lower than 5 µg m$^{-3}$ are rarely observed during the dry season (Pereira et al., 2017 - Atmos. Chem. Phys., 17, 11943–11969; de Miranda, et al., 2012 - Air Qual Atmos Health 5, 63–77; Vasconcellos et al., 2010 - Sci. Total Environ., 408, 5836–5844). Additionally, the sampling site is situated in a green area in the vicinity of areas with heavy traffic, and also there is a local traffic of cars and buses inside the campus. Before the entrance of this cold front, $PM_{2.5}$ levels were high (above 20 µg m$^{-3}$), and a peak in biomass burning contributions was noted (the dark rain event is marked with a purple box in the revised Figure 7).

**C9:** Temperature and RH Time Series: Could you include the time series for temperature and relative humidity during the measurement period in the SI? The averages are mentioned somewhere in the text but the information is lost.

**A9:** Temperature, RH, pressure and irradiance are presented in Figure S1. It was not cited in the manuscript, but it is now on lines 234-236.

**C10:** Figure 6: Could you clarify what is meant by "reference runs" and "BS"?

**A10:** The reference run is the main result obtained in the PMF analysis (Base Model Run). BS (sd) refers to bootstrap runs and their standard deviation, which are performed to evaluate the solution's stability. The caption in Figure 6 was corrected (lines 507-508).

**C11:** PMF Analysis, Section 3.4:

What is the correlation between the identified factors? For example, does VE1 increase after VE2? To support the resuspension? Providing more details on these relationships would enhance understanding.

**A11:** The authors thank the reviewer for the observation. No correlation was observed between the VE1 and VE2 factors. The VE1 is a rather mixed factor (vehicular exhaust plus road dust). It tends to increase with strong, typically drier, northwestern winds (lines 535-536), which facilitate dust resuspension and transport of vehicular exhaust from busy expressways located north/northwest of the site (now marked on the map). This factor presented a negative correlation with relative humidity (RH) and a positive with temperature, indicating that its levels rise under specific meteorological conditions (Section 3.4). The hybrid character of VE1 may result from the combination of urban aerosols—which include vehicle-related pollutants, road dust, and even construction-related emissions—reaching this semi-background site (discussed in Section 3.3). In contrast, the VE2 increases with low wind speeds, suggesting it is related to local vehicular emissions (lines 547-548).

Increasing the time resolution for Positive Matrix Factorization (PMF) can help reduce mixing in source profiles (lines 543-545).

**C12:** Section 3.7 Placement: Consider moving Section 3.7 after section 3.4, as this may improve the manuscript's flow.

**A12:** The authors thank the reviewer for the suggestion, the former section 3.7 was moved after section 3.3 (section numbering was corrected), in lines 622-682.

**C13:** Global Comparison of PMF Analysis: While PMF analysis on $PM_{2.5}$ is discussed for MASP, could you expand on comparisons with findings from other global studies? For example, comparisons to Srivastava et al. (2021), Wang et al. (2018), Han et al. (2023), Cheong et al. (2024), Nava et al. (2020), and others could provide broader insights.

**A13:** Comparisons were included in the paragraphs in section 3.3. Han et al. (2023), Srivastava et al. (2021), Cheong et al. (2024), Nava et al. (2020) were inserted in the discussions about biomass burning contribution to $PM_{2.5}$. Other references were also included, Parra et al. (2024 - Air Qual Atmos Health 17, 599–620), Knorr et al. (2017 - Atmos. Chem. Phys. 17, 9223–9236) and Buchholz et al. (2022 - Nat Commun 13, 2043). They are on lines 494-495 and lines 572-586.

Wang et al. (2018) was included in the discussions on lines 543-544.

Some Brazilian studies were also included in the discussions about sea spray contribution, industrial and vehicular emissions on lines 592-597 and 611-612 (dos Santos et al., 2014 - J. Air Waste Manag. Assoc., 64(5), 519–528; Justo et al., 2023 – Air Qual. Atmos. Health., 16, 289–309; Galvão et al., 2019 - Sci. Total Environ., 651, 1332-1343).

In Section 3.2, Scaramboni et al (2024 – Atmos. Res. 305, 107423) was cited in lines 420-422:

**C14:** Abstract and Figure 7 Discrepancy: In the abstract, you state that "sources related to vehicular emissions remain dominant (over 60% of $PM_{2.5}$). However, Figure 7 shows VE1 and VE2 contributing approximately 40%. Could you clarify why secondary formation (SF) factor is included with car-related emissions? Additionally, Line 452 relates this SF factor to industrial emissions rather than vehicular ones. Moreover, in Section 3.7, the SF factor is described as having a weak correlation with $NO_3^-/EC$ and an opposite trend to the BB, VE1, and VE2 factors. Please reconcile these points.

**A14:** The abstract has been revised to reflect that the contribution has been corrected to 40%. In the previous study from 2014 (Pereira et al., 2017 - Atmos. Environ., 41, 7837–7850), the significance of the sulfate factor (SF) was primarily attributed to vehicle-related emissions. However, since 2019, industrial contributions have become more prominent with the reduction of $SO_2$ emissions (a precursor to secondary sulfate) from vehicles. The CETESB inventories indicate that stationary sources produced five times more $SO_2$ than vehicular sources in the MASP in 2019 (1). More recently, in 2023, CETESB estimated that the industrial sector contributes to more than 90% of $SO_2$ emissions (2) (Figure below; translated from the original taken from the report). In 2000, vehicular emissions still accounted for more than 50% of $SO_2$. It is important to note that most industries are located on the outskirts, whereas vehicular emissions are more concentrated in the city's

central areas. In summary, industrial and vehicular emissions contribute to sulfate formation, and analyzing this distribution could be the subject of further studies. Discussions were added between lines 516-520, in Section 3.3.

The SF factor correlates with $NO_3^-/EC$, particularly due to the enhancement of nitrate on days dominated by SF. Conversely, during periods characterized by primary emissions (BB, VE1, and VE2), the enhancement of EC may reduce this ratio.

The figure below contains the relative emissions by source type in the MASP (extracted from CETESB report), and will not be included in the manuscript or supplementary material):

[Figure]

**A16:** In the current study, it was not possible to differentiate between the two types of vehicles, especially given the 24-hour time resolution used. VE1 is a vehicular exhaust source mixed with road dust, as indicated by the presence of calcium ($Ca^{2+}$) and magnesium ($Mg^{2+}$) ions. VE1 increased with north and northwest winds, suggesting an influence of busy expressways (traffic of passenger cars and trucks). In contrast, the VE2 vehicular source appears to be more localized, as suggested by polar plots, and is less mixed; these local emissions are a mixture of passenger cars and buses.

The markers utilized for urban sampling in the study by Vieira et al. (2023) may have facilitated the separation of these vehicle contributions, as they included additional species in their analysis. Furthermore, their study incorporated black carbon (BC) and elements determined by Energy Dispersive X-ray Fluorescence (EDXRF), including sulfur (S) and phosphorus (P), alongside lead (Pb), which were used to characterize heavy-duty vehicles (HDVs). The sampling site in the East Zone of São Paulo was positioned near busy highways, where a higher proportion of heavy trucks is expected to pass. The presence of an industrial area in that location could also explain the circulation of HDVs. The greater proportion of HDVs, along with their daily and weekly variations, likely contributed to the ability to differentiate this factor.

**C17:** Marine-Related Factor and 6-Factor Solution: On August 15 and 29, a sea breeze was observed. Could a marine-related factor be present? Additionally, would a 6-factor solution help better deconvolute the sources? Some factors in Section 3.4 appear to be mixtures of two or more sources, which a 6-factor approach might resolve. Comparison to global studies will shed light on this study's findings.

**A17:** The authors thank the reviewer for the observation. The sea-spray contribution was relatively low in this period. While we might expect to see an increase in $Na^+$ and $Cl^-$ concentrations during sea breeze events, overall, the impact of these aerosols was minimal, especially for this time of year. This observation aligns with the trajectory clusters discussed earlier. Sea breezes are more frequent during the warmer months (Oliveira et al., Water, Air, & Soil Pollution: Focus 3, 3–15, 2003), whereas this study was conducted in autumn and winter. We found low correlations between $Na^+$ and $Cl^-$, as discussed in the excerpt below (in lines 673-676):

"$Na^+$, as previously observed in the MASP (Vieira-Filho et al., 2016b), was not specific for any source and presented a weak correlation with some other species ($Ca^{2+}$, $Cl^-$, $PO_4^{3-}$, As, Rb, Sr, and Cd) (r > 0.3), without strong correlation with $Cl^-$, which can be explained by the relatively low influence of sea spray in this site in the studied period"

We opted for the five-factor solution because it yielded the best results. The six-factor solution produced a factor solely for copper, but its bootstrap value was only 71%, which we considered unsatisfactory. In contrast, the five-factor solution offered better bootstrap results (above 87%), making it more reliable.

Factor 6 (only copper), from the rejected six-factor solution:

[Figure]

**C18:** Pollution Events and Particle Size Distribution: This section could benefit from revision to provide greater depth. The first part reads like an introductory paragraph, while the second discusses four specific days without adequately explaining their significance. Additionally, the entire measurement period is only briefly addressed. Including a table summarizing average values (e.g., new particle formation rates, growth rates, nucleation event days) in the main manuscript would enhance clarity and aid the reader.

**A18:** The section was reformulated, providing more information about the entire measurement period, and providing context for the specific days presented in Figure 9.

Lines 108-109 - Phrase reformulation: PNSD measurements were taken between June to September 2019, totalizing 48 non-consecutive sampling days during the austral winter.

Section 3.5 (corrected) title was reformulated: Particle size distributions: associations with aerosol source apportionment and meteorological scenarios

Reformulated section (lines 685-765):

[revised manuscript text omitted]

**C19:** Enrichment Factor (EF) Results: Although the methodology for calculating enrichment factors is described in the data treatment section, no corresponding results, tables, or figures are provided in the main text. Apart from the brief mention in Lines 610–611, could you include these results to strengthen the manuscript?

**A19:** The figure was moved from SI to the manuscript. The discussion was improved on lines 809-822:

"Enrichment factors for the elements were above 10 for Cr, Cu, Zn, As, Se, Mo, Ag, Cd, Sn, Sb, Ba, Tl, Pb, Bi, and U (Figure 11), suggesting a strong anthropogenic character. Despite the very low concentrations, Se, Sb, and Bi presented the highest EFs, indicating a very high anthropogenic character. Selenium was previously attributed to anthropogenic sources such as fossil fuel combustion, burning of garbage, tires and paper, coal combustion, oil, and glass industries (Mehdi et al., 2013), and previously attributed to industrial sources in the eastern part of the MASP (Vieira et al., 2023). Antimony is associated with non-exhaust emissions (brake wear) (Thorpe and Harrison, 2008) and associated with non-ferrous metal industries (Calvo et al., 2013). Bismuth is related to anthropogenic sources, such as fossil fuel combustion, metallurgy, and refuse incineration (Ferrari et al., 2000). Other species presented low EFs, being attributed to geogenic sources: Ti, Fe, and Co (EFs near and below 1). Often associated with catalyst emissions (Kulkarni et al., 2006), La and Ce displayed relatively low enrichment factors (below 5)."

Minor Comments:

**C20:** Line 89. Replace the word "extended"

**A20:** The authors thank the reviewer for the comment, it was replaced (line 90).

**C21:** Line 118. Rewrite the phrase "the first…derivatives". The second step looks like is missing something.

**A21:** The phrase was reformulated (lines 122-124).

**C22:** Lines 157-159. Replace "BaPTEQ" to "BaP$_{TEQ}$", "BaPMEQ" to "BaP$_{MEQ}$", "BaPEq" to "BaP$_{Eq}$". Fix/increase the "(2)" to be the same as the others. Explain in what units the BaA, BkF etc. are referring to.

**A22:** We have corrected the subscripts; however, there may be an issue when converting document files to PDF. This will be addressed in the final editing process. The concentrations refer to the species, and this was updated on line 164. The acronyms for the PAHs are listed in section 2.2.

**C23:** Line 177. Add reference after "depleted". Replace "It" with "EF". Add "with" before EF.

**A23:** A reference was added (line 187). The text was reformulated (lines 186-188).

**C24:** Lines 179-180. Either use parentheses for all CXp etc., or for none of the four. I recommend you use parentheses for none. Also, from which study do you take the concentration in the Earth's crustal material? Please add references.

**A24:** This information was taken from the Appendix of Lee's (1999) textbook "Concise Inorganic Chemistry, 5th Edition" (Appendix A. Abundance of the elements in the earth's crust), this is described in lines 187-188. Parentheses were removed.

**C25:** Line 251. Please present Figure 2 in order, for example not first the Fig. 2e and then Fig. 2c-d. Change the order of chart pies in Figure 2 to be consistent with the manuscript.

**A25:** Figure 2 was completely edited. Now, $PM_{2.5}$, OM, and EC are on item (a), Lev and Lev/Man are on item (b), mass balance is on item (c), ion composition is on item (d), OC composition is on item (e), and EC composition is on item (f).

[Figure]

**C26:** Lines 252-256. What are EC1, EC2, OC1, OC2 etc. are referring to? Please explain.

**A26:** The thermal-optical analysis was performed under EUSAAR thermal protocols. In the thermal optical analysis, nine temperature-resolved carbon fractions are obtained: four organic carbons (OC1, OC2, OC3, and OC4), four elemental carbons (EC1, EC2, EC3 and EC4) and a organic pyrolyzed carbon fraction (PC) is also obtained. The first stage in helium medium is carried out under the following conditions of temperature and time steps: 200 °C for 120s (OC1), 300 °C for 150s (OC2), 450 °C for 180s (OC3) and 650 °C for 180s (OC4). The second stage, under He-O2 medium, the four steps are: 500 °C for 120s

(EC1), 550 °C for 120s (EC2), 700 °C for 70s (EC3) and 850 °C for 80s (EC4). This was better explained in lines 141-144.

**C27:** Figure 2. Increase the fonts for Lev, Lev/Man, PM2.5, EC and OM. Add to the caption that the time series in a and b refer to daily data. What do you mean by "dark" rain event?

**A27:** The fonts in Figure 2 were increased. The dark rain is now described in the methodology (lines 91-91).

**C28:** Line 276. Add reference of previous campaign (2014).

**A28:** It was added to the text (line 305).

**C29:** Figure 4. Increase fonts. Increase the graph to be clearer to the reader. It would be nice to add the annual limit recommended by the European Environment Agency with different color.

**A29:** Figure 4 has been updated to include a line indicating the EEA limit:

[Figure]

**C30:** Figure 5: Could you provide additional details in the main text on how these polar plots were generated?

**A30:** The polar plot is obtained by a function in the OpenAir package (RStudio), which plots pollutant concentration in polar coordinates, showing concentration by wind speed and direction. More information was added to the manuscript (lines 174-175).

**C31:** Line 362. The number "10" and "9" it is not very clear to what they are referring to.

Line 364. Same for number "19"

**A31:** It refers to Lev/Man ratio, it was added to the text (lines 413-415).

**C32:** Line 367. Maybe wanted to write "Figure 2" instead of "Figure 1"?

**A32:** Yes, Figure 2. It was corrected (line 418).

**C33:** Line 434. Mention the factors by the order you present those in Figure 6. It's easier for the reader.

**A33:** The phrase was reformulated (lines 487-489).

**C34:** Figure 6. Increase all fonts (axis, species, Factors). Especially, the "y-axis" cannot be read. Also, you could use a common x-axis in the bottom of the figure and increase the bars (maybe use different colors for each factor).

**A34:** Figure 6 was revised and inserted in the manuscript:

[Figure]

**C35:** Line 457. Move "Factor 3" to another paragraph.

**A35:** The Factor 3 discussions were moved to another paragraph (line 521).

**C36:** Figure 7. Better use µg m-3 instead of ng m-3. Increase fonts. Add dark rain event purple bar.

**A36:** Changes were performed in the figure:

[Figure]

**C37:** Figure 8. In the caption there is a "b)" in the end of the sentence without anything following. Maybe move a) the b) before "trajectory" and "participation", respectively.

**A37:** (a) refers to "Trajectory analysis for the contribution of factor 3" and (b) to "participation of industries for municipalities in São Paulo state (industrial transformation value)" (lines 619-620).

**C38:** Line 541. Remove the double dot.

**A38:** The double dot was removed.

**C39:** Figure 9. Increase all fonts. At the caption, move (a)-(d) before the dates.

**A39:** Figure and captions were revised:

[Figure]

**C40:** Line 561. Units for number concentration?

**A40:** The units for particle number concentration were included in the referred sentence, cm$^{-3}$. We thank the reviewer for the correction.

**C41:** Lines 562-563. "The distribution, which had a geometric mean diameter of 94.7 ± 2.0 nm at 8 AM, shifted to particles with a mean size of 43.8 ± 2.2 nm." And? What is the explanation for this? The key point?

**A41:** The section was reformulated, improving the discussion about the impacts of meteorological phenomena on the dynamics of particle number size distributions.

**C42:** Line 570. A dot is missing after "…83%)"

**A42:** The dot was inserted (line 769).

**C43:** Section 3.6: A Table with the polluted/ clean concentration values for all species would be an interesting addition.

**A43:** A table comparing average concentrations in polluted and clean was created (Table S6) and cited in the text (line 781).

**C44:** References:

Cheong, Y., Kim, T., Ryu, J. *et al.* Source apportionment of PM2.5 using DN-PMF in three megacities in South Korea. *Air Qual Atmos Health* (2024). https://doi.org/10.1007/s11869-024-01584-5.

Yun-Sung Han, Da-Mee Eun, Greem Lee, Sung Yong Gong, Jong-Sang Youn, Enhancement of PM2.5 source appointment in a large industrial city of Korea by applying the elemental carbon tracer method for positive matrix factorization (PMF) model, Atmospheric Pollution Research, Volume 14, Issue 11, 2023, 101910, ISSN 1309-1042, https://doi.org/10.1016/j.apr.2023.101910.

Nava, S.; Calzolai, G.; Chiari, M.; Giannoni, M.; Giardi, F.; Becagli, S.; Severi, M.; Traversi, R.; Lucarelli, F. Source Apportionment of PM2.5 in Florence (Italy) by PMF Analysis of Aerosol Composition Records. *Atmosphere* 2020, *11*, 484. https://doi.org/10.3390/atmos11050484.

Srivastava, D., Xu, J., Vu, T. V., Liu, D., Li, L., Fu, P., Hou, S., Moreno Palmerola, N., Shi, Z., and Harrison, R. M.: Insight into PM2.5 sources by applying positive matrix factorization (PMF) at urban and rural sites of Beijing, Atmos. Chem. Phys., 21, 14703–14724, https://doi.org/10.5194/acp-21-14703-2021, 2021.

Vieira, E. V. R., do Rosario, N. E., Yamasoe, M. A., Morais, F. G., Martinez, P. J. P., Landulfo, E. and Maura de Miranda, R.: Chemical characterization and optical properties of the aerosol in São Paulo, Brazil, Atmosphere, 14(9), 1460, https://doi.org/10.3390/atmos14091460, 2023.

Wang Q.Q., L.P. Qiao, M. Zhou, S.H. Zhu, S. Griffith, L. Li, J.Z. Yu Source Apportionment of PM2.5 Using Hourly Measurements of Elemental Tracers and Major Constituents in an Urban Environment: investigation of Time-Resolution Influence J. Geophys. Res.-Atmos., 123 (2018), pp. 5284-5300.

Zhao, J., Zhang, Y., Xu, H., Tao, S., Wang, R., Yu, Q., Chen, Y., Zou, Z., Ma, W., 2021. Trace elements from ocean-going vessels in east Asia: vanadium and nickel emissions and their impacts on air quality. J. Geophys. Res. Atmos. 126, 1–16. https://doi.org/ 10.1029/2020JD033984.

**A44:** References were included in the text.

---

## Author Response (AR2)

Dear Editor,

The final changes have been made to the manuscript, as suggested. Several paragraphs were rewritten for clarity, and punctuation was improved. Citations to Farias et al. were removed since that article has not yet been accepted (lines 372-374, 377-378, 603-605, and 610-612). Additionally, the figures were edited to enhance resolution and ensure uniformity (zipped file). The tracked version of the manuscript is attached (below); lines refer to this version.

-        Minor punctuation and other grammar, and word choice adjustments were made throughout the text (lines 37, 39, 42, 44, 45, 49, 51, 52, 54, 55, 60, 62-65, 72, 74, 89, 92, 100, 101, 121, 128, 153, 155, 156, 184, 185, 192, 197, 198, 199, 203, 204, 208, 217, 218, 224, 244, 245, 247, 256, 269, 273, 278, 282, 283, 290, 307-310, 311, 312, 323-333, 336-342, 344, 345, 350, 351, 360-362, 365-368, 381, 382, 384, 385, 386, 396, 397, 399-405, 408, 409, 412, 415-425, 436, 445, 448, 455, 461-463, 475, 480-482, 484, 485, 488, 490, 491, 495, 496, 503-506, 506, 516, 519-521, 523, 527, 531-534, 538-540, 544-545, 547-550, 553, 554, 556-558, 561, 567-573, 575-578, 581, 582, 591, 592, 594-596, 599, 604, 609-611, 629-631, 632, 637, 639, 640, 643-649, 651-657, 667-668, 677, 679-684, 686-688, 695, 706, 727, 728, 741, 749, 764, 765, 767, 768, 770, 772, 774, 785, 786, 788, 790, 792, 795, and 843).

-        A lost phrase was removed from line 40.

-        Direct and indirect climatic effects of PM were highlighted in line 44.

-        Some paragraphs received major revisions (Introduction, Sections 3.2, 3.3, and 3.4). Some phrases were reordered for improved fluidity and coherence in the paragraphs in lines 46-58, 59-82, 381-388, 591-618, and 666-676.

-        Figure 2 caption was revised for clarity (lines 260-261).

-        A phrase was added to complete the discussions in lines 647-648

-        The reference to the countryside of São Paulo was corrected to its proper geographic location (north and northwest) on lines 70-71 and 325-326.

-        References were added on lines 339, 608, and 648.

[revised manuscript text omitted]